# Targeting of eIF6-driven translation induces a metabolic rewiring that reduces NAFLD and the consequent evolution to hepatocellular carcinoma

Alessandra Scagliola[1,9], Annarita Miluzio[1,9], Gabriele Ventura[2], Stefania Oliveto[1], Chiara Cordiglieri [1], Nicola Manfrini [1,2], Delia Cirino[2], Sara Ricciardi[1,2], Luca Valenti[3,4], Guido Baselli[3], Roberta D'Ambrosio[5], Marco Maggioni[6], Daniela Brina[7], Alberto Bresciani [8] & Stefano Biffo [1,2✉]

A postprandial increase of translation mediated by eukaryotic Initiation Factor 6 (eIF6) occurs in the liver. Its contribution to steatosis and disease is unknown. In this study we address whether eIF6-driven translation contributes to disease progression. eIF6 levels increase throughout the progression from Non-Alcoholic Fatty Liver Disease (NAFLD) to hepatocellular carcinoma. Reduction of eIF6 levels protects the liver from disease progression. eIF6 depletion blunts lipid accumulation, increases fatty acid oxidation (FAO) and reduces oncogenic transformation in vitro. In addition, eIF6 depletion delays the progression from NAFLD to hepatocellular carcinoma, in vivo. Mechanistically, eIF6 depletion reduces the translation of transcription factor C/EBPβ, leading to a drop in biomarkers associated with NAFLD progression to hepatocellular carcinoma and preserves mitochondrial respiration due to the maintenance of an alternative mTORC1-eIF4F translational branch that increases the expression of transcription factor YY1. We provide proof-of-concept that in vitro pharmacological inhibition of eIF6 activity recapitulates the protective effects of eIF6 depletion. We hypothesize the existence of a targetable, evolutionarily conserved translation circuit optimized for lipid accumulation and tumor progression.

[1] Istituto Nazionale di Genetica Molecolare, INGM, "Romeo ed Enrica Invernizzi", Milan, Italy. [2] Department of Biosciences, University of Milan, Milan, Italy. [3] Department of Pathophysiology and Transplantation, University of Milan, Milan, Italy. [4] Translational Medicine, Department of Transfusion Medicine and Hematology, Fondazione IRCCS Ca' Granda Ospedale Policlinico, Milan, Italy. [5] Department of Hepatology, Fondazione IRCCS Ca' Granda Granda Ospedale Policlinico, Milan, Italy. [6] Department of Pathology, Fondazione IRCCS Ca' Granda Ospedale Policlinico, Milan, Italy. [7] Institute of Oncology Research, Oncology Institute of Southern Switzerland, Università della Svizzera Italiana, Bellinzona, Switzerland. [8] Department of Translational and Discovery Research, IRBM S.p.A., Pomezia (Roma), Italy. [9] These authors contributed equally: Alessandra Scagliola, Annarita Miluzio. ✉email: biffo@ingm.org

The accumulation of lipids in the liver is known as nonalcoholic fatty liver disease (NAFLD)[1]. The potential evolution of NAFLD to cirrhosis and hepatocellular carcinoma (HCC) makes NAFLD of clinical importance[2]. Nonalcoholic steatohepatitis (NASH), the progressive form of NAFLD, is a risk factor for the onset of HCC. The annual progression rate of cirrhotic NASH to HCC ranges from 2.4% to 12.8%[3]. Importantly, these pathologies are interconnected and can be ascribed to a disorder triggered by the fact that lipid accumulation exceeds lipid consumption, generating a chronic liver insult that may lead to fibrosis and hepatic failure[1]. To overcome the evolution of NASH, lipid metabolic enzymes including acetyl-CoA carboxylase, a key enzyme in the commitment to fatty-acid synthesis (FAS)[4], have been pharmacologically targeted, but the fat reduction in the liver is accompanied by increased circulating triglycerides. A better strategy for preventing the evolution from NAFLD to HCC might be to change the global adaptation of the metabolic machinery, rather than the activity of a single enzyme. In this context, evolutionarily conserved nutrient-sensing pathways[5] have formed under the pressure to optimize energy storage in conditions of sporadic food intake. Consequently, these evolutionarily selected pathways maladapt to chronic, increased nutrient supply. We speculate that if specific nodes of nutrient-sensing pathways can be manipulated, then they can steer toward reducing, rather than increasing, the negative effects elicited by an increased nutrient load. This strategy, in turn, may prevent the transition from NASH to HCC.

Insulin signaling is a nutrient-sensing pathway, conserved from Drosophila to humans. One effect of insulin, across kingdoms and cells, is the stimulation of protein synthesis, i.e., translation[6]. The liver is the adult organ with the highest rate of protein synthesis[7]. Within minutes from the increase of blood nutrients, the rate of translation in the liver increases[8], raising intriguing questions. If an increase of protein synthesis stimulated by oncogenic pathways can be easily explained by the need of cell growth, what is the role of increased protein synthesis in the liver, in the absence of conspicuous proliferation? Also, it is often neglected that insulin and oncogenic pathways stimulate protein synthesis converging on identical translation factors[9]. What is the common process regulated by translation, shared by cycling cells and post-mitotic hepatocytes?

In the past, we identified eukaryotic initiation factor 6 (eIF6) as a translation factor that is rate-limiting for the full stimulation of protein synthesis, driven by either insulin or growth factors[10,11]. Knockout mice with eIF6 haploinsufficiency are healthy, but resistant to oncogenic transformation[12,13]. Serendipitously, we found that eIF6 knockout mice had also reduced white fat[12]. We demonstrated that eIF6 activity potentiates the translational efficiency of mRNAs encoding for lipogenic transcription factors that contain regulatory uORFs in their 5′ UTR[14]. The previous observations raised the possibility that interfering with eIF6 activity would interrupt the translational reinforcement of de novo lipogenesis, with a strong impact on all aspects of the evolution from obesity to HCC. Here, we unveil the existence of an evolutionarily conserved translation circuit optimized for lipid accumulation and predisposing to tumor progression and provide proof-of-concept that the manipulation of eIF6 can be exploited for preventing liver disease progression.

## Results

### Higher eIF6 levels mark the progression from NAFLD to HCC, independently from other translation machinery factors.
We asked whether during the progression from normal liver to NAFLD, the levels of eIF6 increase, suggesting a translational amplification of the lipogenic program. To approach the problem, we first analyzed a data set of human patients[15]. We used a matrix of mRNAs belonging to the protein synthesis machinery,[16] that represent a critical step of translational control, and assessed their levels. We found that during NAFLD progression eIF6 mRNA levels increase (Fig. 1a). In contrast, mRNAs encoding for structural proteins of the small (rpSs, rack1) and large (rpLs) ribosomal subunits, or for selected eIFs decrease during NAFLD progression (Fig. 1a). These data rule out a general amplification of the translational apparatus. Next, we used an in-house collected cohort[17], and performed an immunohistochemistry analysis of eIF6 and of phosphorylated ribosomal protein S6 (rpS6). Phosphorylated rpS6 (Ser235/236) is a proxy for the activation of the mTORc1 pathway[18]. We found a negative correlation between the progression of liver disease and the phosphorylation of rpS6 (Fig. 1b). Instead, eIF6 protein levels increase during disease progression, both in the cytoplasm and in the nucleoli (Fig. 1c). Quantitative measurements of phosphorylated rpS6 and eIF6 signals confirmed the histological observation (Fig. 1d).

eIF6 is an accelerator of tumorigenesis and tumor progression[13,19–21]. Cirrhosis represents a risk for HCC. Using an available single-cell RNA-seq data set, we found that eIF6 mRNA expression levels are higher in two cell lineages derived from cirrhotic livers[22], cholangiocytes, and hepatocytes (Fig. 1e). Moreover, survival analysis showed that higher eIF6 mRNA levels trend with decreased overall survival[23] in primary liver cancer (Fig. 1f). Last, primary tumors of metastatic patients[24] have higher eIF6 mRNA levels than primary tumors without metastatic involvement (Fig. 1g). Together, these findings show that there is a positive correlation between eIF6 expression, lipid accumulation, and the progression of liver disease. Next, we tested whether a cause-effect relationship exists.

### eIF6 haploinsufficiency reduces cellular transformation, inhibits intracellular lipid synthesis, and increases FAO.
Depletion of eIF6 reduces tumorigenesis upon Myc-induced lymphomagenesis[13], and reduces the accumulation of white fat[12]. We, therefore, analyzed whether the effects on cellular transformation and lipid accumulation are cell-autonomous and coexist, in vitro. First, we performed a transformation assay on both eIF6-expressing and eIF6-silenced AML12 hepatocytes using a retroviral vector carrying DNp53 plus oncogenic H-ras^V12. eIF6-depleted AML12 hepatocytes transformed at lower efficiency than control ones (Fig. 2a) and formed smaller transformed colonies (Fig. 2b), normalized on equal H-ras^V12 transduction (Fig. 2c).

Next, we analyzed de novo FAS and lipid accumulation. In order to analyze FAS, we provided $^{14}$C-glucose and analyzed $^{14}$C incorporation in fatty acids. In order to analyze lipid accumulation from exogenous sources, we cultured cells in the presence of palmitate and measured Oil Red O staining. We found that eIF6 depletion, in AML12, reduces both FAS (Fig. 2d, Supplementary Fig. 1a) and intracellular lipid content (Fig. 2e, f; Supplementary Fig. 1b). Since an increase of fatty-acid oxidation (FAO) may contribute to the reduction of lipid accumulation, we analyzed FAO rates by measuring $^{14}CO_2$ release upon administration of exogenous $^{14}$C-palmitate. We found that eIF6-depleted AML12 hepatocytes had increased oxidation of palmitate that could be reduced by inhibition of mitochondrial FAO by Etomoxir (Fig. 2g). This result was also confirmed ex vivo: liver extracts from eIF6 heterozygous (eIF6$^{+/-}$) mice showed increased $^{14}CO_2$ production compared with the wild-type (eIF6$^{+/+}$) counterpart (Fig. 2h).

Last, we asked whether re-expression of eIF6 in a model depleted by eIF6, was able to induce lipid accumulation. To perform this experiment, we used eIF6$^{+/-}$ Ear mesenchymal stem cell culture (EMSC) differentiated to adipocytes. Re-expression of eIF6 in eIF6$^{+/-}$ EMSC adipocytes, restored lipid accumulation (Supplementary Fig. 1c) demonstrating that the effects of eIF6 are

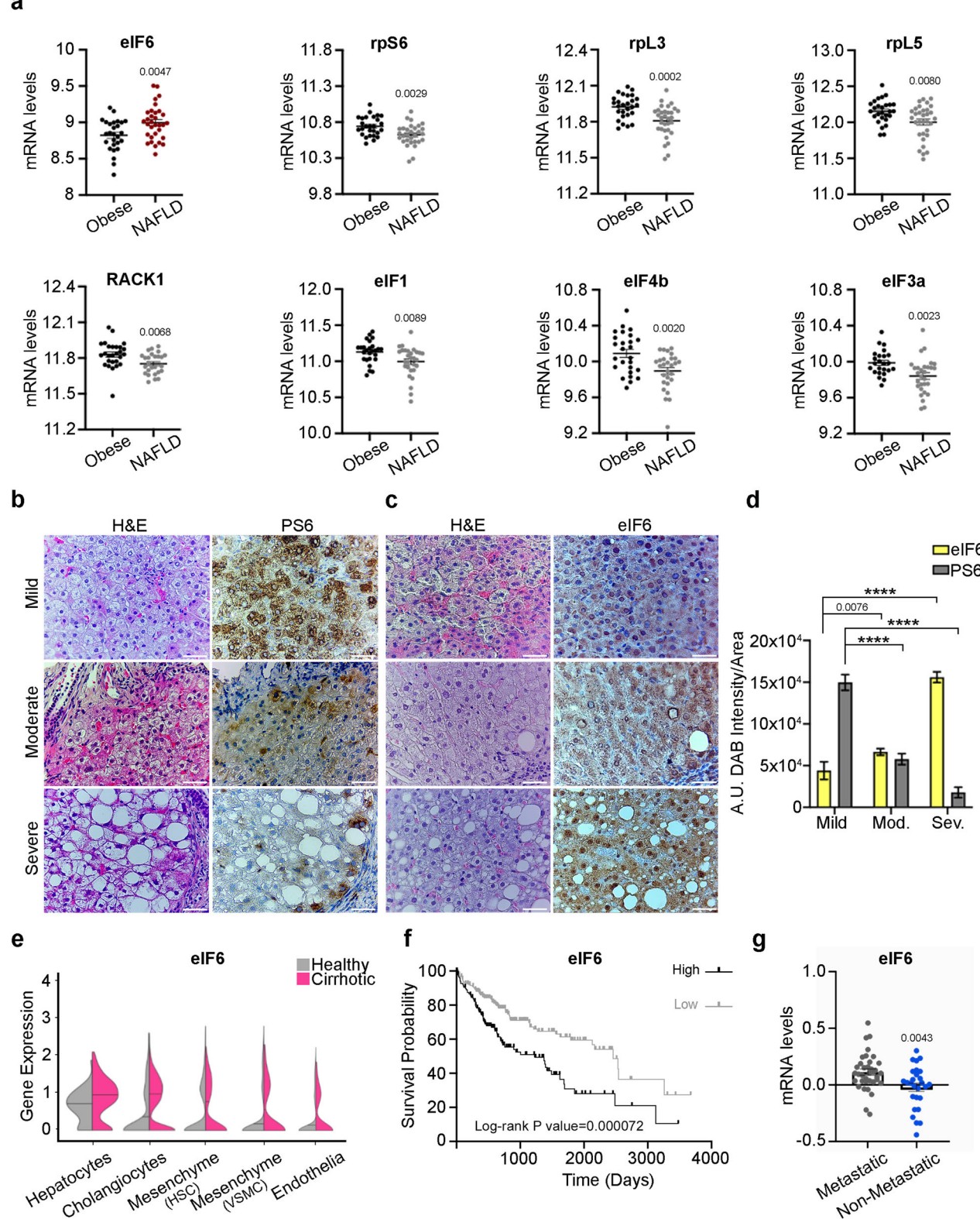

reversible. These data corroborate the hypothesis of a causal relationship between eIF6 dosage, intracellular lipid metabolism, and cellular transformation. We moved to an in vivo model.

**eIF6 depletion protects from HFD-induced obesity and insulin resistance, and reduces HFD-induced hepatic steatosis and fibrosis progression**. We asked whether eIF6 haploinsufficiency

could affect the onset and the progression of NAFLD, in vivo. We opted for a high-fat diet (HFD) containing 45% calories derived from fat. We compared eIF6 heterozygous (eIF6$^{+/-}$) mice having half of eIF6 protein levels to matched eIF6 wild-type (eIF6$^{+/+}$) mice[12]. During 16 weeks of chronic HFD feeding, the increase in body weight of eIF6$^{+/+}$ mice was greater than that of heterozygous ones (Fig. 3a, b), with no change in daily caloric intake

**Fig. 1 Progression of human NAFLD to cirrhosis and HCC correlates with an increase of eIF6 that counteracts a drop of ribosomal proteins. a** Data-mining studies, in NAFLD patients, reveal an increase in eukaryotic Initiation Factor 6 mRNA and a concomitant reduction of expression of genes encoding for ribosomal proteins (rpL; rpS) and initiation factors (eIFs). Two-tailed *t* test. Data are presented as mean ± SEM. **b**, **c** Representative H&E and IHC staining on human liver sections show a reduction of rpS6 phosphorylation (left) and overexpression of eIF6 (right) during NAFLD progression. Scale bars = 50 µm. **d** Quantification of eIF6 and PS6 (phosphorylated rpS6) signals in mild, moderate, and severe steatosis. Data are presented as mean ± SD. ****p value ≤ 0.0001. N = 12/class. Two-tailed *t* test. **e** Violin plots showing the expression levels of eIF6 in healthy (gray) and cirrhotic (purple) livers. Black lines denote the median. Retrieved from single-cell RNA-seq[22]. **f** KM survival curves of primary liver cancer patients stratified for eIF6 mRNA levels. Data are retrieved from The Human Pathology Atlas[54]. Log-rank *P* value was used for statistical significance. Hazard ratio = 1.970; 95% confidence interval of ratio = 1.374 to 2.824. **g** Expression levels of eIF6 in primary tumors of metastatic liver cancer, compared with not metastatic. Data are presented as mean ± SEM. Two-tailed *t* test. *NAFLD* nonalcoholic fatty liver disease, *HCC* hepatocellular carcinoma. Data are provided as a Source Data File.

(Supplementary Fig. 2a). Plasma triglycerides and cholesterol levels were lower in eIF6$^{+/-}$ mice (Fig. 3c). Plasma glucose levels were identical in the two cohorts (Supplementary Fig. 2b), but eIF6$^{+/-}$ mice were more responsive to ITT, insulin tolerance test (Fig. 3d). Overall, these data confirm that in vivo eIF6 depletion reduces lipid accumulation and better preserves insulin sensitivity.

We addressed the formation of fatty liver formation by analyzing the livers of eIF6$^{+/+}$ and eIF6$^{+/-}$ mice after 110 days of HFD feeding. At autopsy, the pale white color, a sign of increased lipid content, and organ size were attenuated in mice heterozygous for eIF6 (Fig. 3e), confirmed by liver/body weight ratio measurement (Fig. 3f). Histological examination of liver sections showed that eIF6$^{+/-}$ mice displayed less steatosis and less Oil Red O accumulation (Fig. 3g). Sirius Red staining is used for diagnosis of fibrosis, to detect collagens: eIF6$^{+/-}$ livers had less Sirius Red staining (Fig. 3h). These morphological changes were confirmed by quantitative measurements of Oil Red O that indicated the reduced size of lipid droplets (Fig. 3i, j) and Sirius Red staining that indicated reduced fibrosis (Fig. 3k). Collectively, our results demonstrate that 50% eIF6 reduction slows down hepatic steatosis and fibrosis.

**eIF6 inhibition reduces de novo lipogenesis acting on C/EBPβ translation**. We evaluated by RNA-seq the transcriptional changes driven by chronic eIF6 depletion in the liver of mice subjected to HFD (Supplementary Data 1)) and reconstructed a mechanistic model. Transcriptome analysis revealed mRNA differences between eIF6$^{+/+}$ and eIF6$^{+/-}$ livers which support a coordinated, metabolic rewiring (Fig. 4, Supplementary Fig. 3a, b). Specifically, we found that: (i) similarly to what was observed in human samples, ribosomal protein mRNAs show an opposite trend as compared with eIF6 expression. In addition, we found in eIF6$^{+/-}$; (ii) a coordinated downregulation of genes belonging to de novo lipogenesis and to the collagen fibril organization pathway; (iii) an increase in the expression of mitochondrial factors (Fig. 4a and Supplementary Fig. 3b). The variations found by RNA-Seq were confirmed by quantitative real-time PCR (Fig. 4b–d). The effects on gene expression driven by eIF6 were both pervasive and peculiar. In the lipogenic pathway, among downregulated genes, we found *Acc1* and *Fasn* for fatty acids, *Hmgcr* and *Mvd* for cholesterol, *Gpam*, *Mogat1* for glycerolipids, and triacylglycerol. Collagen-encoding genes, *Col3a1* and *Col4a1*, were downregulated in line with reduced fibrosis. eIF6 deficiency caused also the downregulation of *Pparα* and *Pparγ* mRNAs (see discussion). Overall these data suggest that chronic inhibition of eIF6 during HFD preserves translation and mitochondrial integrity reducing fibrosis.

eIF6 increases the translation of uORF-containing mRNAs. One prominent translational target of eIF6 is C/EBPβ, a master factor of adipogenesis and an upstream regulator of PPARγ[25], specifically in the LIP isoform[14]. We analyzed whether C/EBPβ was reduced at the translational level by measuring the levels of C/EBPβ mRNA on polysomes. Although the total level of C/EBPβ

mRNA was slightly increased (Supplementary Fig. 4a), its association with heavy polysomes was reduced (Supplementary Fig. 4b). C/EBPβ-LIP protein was also reduced in eIF6$^{+/-}$ livers, confirming the downregulation of this uORF-encoded protein at the translational level (Supplementary Fig. 4c).

LIP protein induces a pro-carcinogenic metabolic reprogramming[26]. We rescued C/EBPβ-LIP protein levels in eIF6-depleted cells. Re-expression of C/EBPβ-LIP protein restored lipid accumulation (Supplementary Fig. 4d–g). Finally, network analysis confirms that C/EBPβ translational downregulation matches a transcriptional downregulation of its lipogenic targets (Supplementary Fig. 4h), leading to a reduction in de novo lipogenesis. Overall, these data show that eIF6 orchestrates a pro-adipogenic, pro-fibrotic gene expression profile, at least partly through C/EBPβ-LIP.

**eIF6 depletion impairs the precancerous transcriptional signature of the liver and reduces NAFLD to HCC progression**. Given the double effects of eIF6 on lipid synthesis and hepatocyte transformation, in vitro, we asked whether the progression to HCC was reduced by eIF6 haploinsufficiency. Notably, metadata analyses on mice models show that eIF6 mRNA is overexpressed during HCC progression in a statistically significant fashion (Supplementary Table 1). We also asked whether the genes regulated by eIF6 actually belong to the subset of genes associated with tumor progression in human HCC (Supplementary Data 2). Signatures for genes that mark the transition from NASH to HCC in humans have been recently proposed[27,28]. We found considerable overlap between genes deregulated in HCC and altered in eIF6 knockouts, upon HFD. The overlap was seen both in a list of genes with diagnostic and/or prognostic value[28] (Fig. 5a, b; Supplementary Table 2 and Source Data, Excel file Human Data), as well as in a signature of 25 genes that connects dyslipidemia to HCC[27], in which 18 overlaps with the signature of genes inhibited by eIF6 depletion (Supplementary Table 3 and Source Data, Excel file Human Data). Along this line, Gene Ontology functional analysis predicts, as expected by phorbol 12-myristate 13-acetate stimulation of eIF6 activity[10], tyrosine kinase phosphorylation as a major eIF6-regulated oncogenic-related event (Fig. 5c). In conclusion, gene expression analysis suggests that eIF6 knockout mice may be protected from the evolution of NASH to HCC.

To directly test the relationship between eIF6 protein expression and the greater risk of progression to HCC, we addressed, in vivo, the effect of eIF6 depletion on liver cancer development. Both eIF6$^{+/+}$ and eIF6$^{+/-}$ mice were subjected to HFD diet, high sugar drinks water, and weekly CCl$_4$-treatment, whose combination induces the NAFLD rapid progression to steatohepatitis, fibrosis (NASH), and to HCC (Fig. 5d)[29]. After 24 weeks, eIF6$^{+/-}$ mice showed reduced incidence and number of surface tumors, as indicated in Fig. 5e. Quantification of tumors size distribution indicates that eIF6$^{+/-}$ mice had a lower number of surface neoplastic formations, in all considered dimensional ranges (Fig. 5f). Notably, we observed tumors reaching their

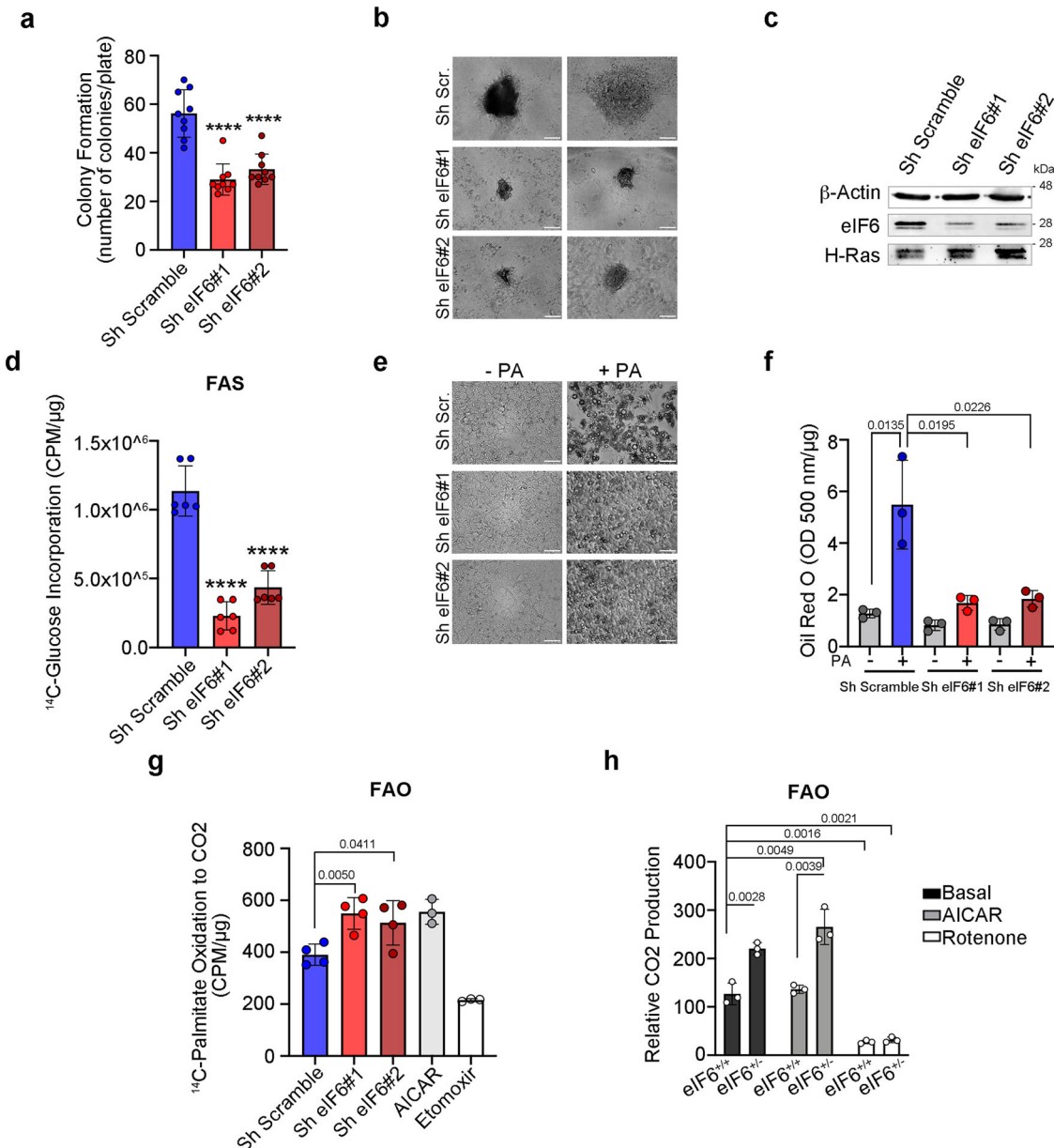

**Fig. 2 Downregulation of eIF6 prevents oncogenic transformation, lipid synthesis, and activates palmitate-stimulated fatty-acid oxidation. a–c** Reduced transformation by eIF6 depletion. Control and eIF6-silenced cells were transduced with DNp53-H-Rasv12 retrovirus: the number of transformed tumor colonies are reduced in eIF6-silenced cells. Data are represented as means ± SD of $n = 9$ plates/condition; ****$p$ value ≤ 0.0001 **a**. Bright-field images of transformed colonies. Scale bars = 100 μm **b**. eIF6 levels in wt and silenced eIF6-transformed cells. Anti H-Ras antibody was used as a control for retroviral infection **c**. **d** FAS (fatty-acid synthesis) analysis on AML12 control and eIF6-silenced cells: eIF6 downregulation reduces $^{14}$C-Glucose incorporation ($n = 6$). Two-tailed $t$ test. ****$p$ value ≤ 0.0001. **e** Representative images of AML12 treated with 2 mM palmitate or ethanol, as control: depletion of eIF6 by shRNA reduces intracellular lipid accumulation. Scale bar = 50 μm. **f** Quantification of Oil Red O staining with or without exogenous palmitate addition. $N = 3$. Lipid droplets accumulation is reduced in eIF6-depleted AML12 cells. Data are represented as means ± SD. Two-tailed $t$ test. ***$p$ value ≤ 0.001. **g** FAO (fatty-acid oxidation) rates were measured on $^{14}$CO$_2$ release upon administration of exogenous $^{14}$C-palmitate. Data are means ± SD ($n \geq 3$). AICAR and Etomoxir were used as positive and negative controls, respectively. Two-tailed $t$ test. **h** FAO rates in isolated livers from WT and eIF6$^{+/-}$ mice either in the presence of AICAR or rotenone. $N = 3$. Two-tailed $t$ test. Data are means ± SD. Data are provided as a Source Data File.

maximal size, >10 mm in diameter, only in eIF6$^{+/+}$ mice (Fig. 5f). Morphological observation of eIF6 heterozygous liver sections showed that they had smaller neoplastic regions surrounded by extensive areas of steatosis compared to wt ones (Fig. 5g). Morphometric analysis of the tumor mass revealed a higher percentage of small nodules in eIF6$^{+/-}$ mice (Fig. 5h), and a reduction of eIF6 protein levels (Fig. 5i). Finally, we evaluated the expression of specific genes involved in NASH transition to HCC: genes associated with lipid storage processes and liver

tumorigenesis were maintained at lower levels in eIF6$^{+/-}$ mice (Fig. 5j). Taken together, these findings demonstrate that 50% eIF6 depletion protects mice from diet-induced obesity and diminishes the progression to aggressive liver cancer.

**Mitochondrial fitness is preserved in eIF6$^{+/-}$ primary hepatocytes owing to the maintenance of an active alternative pattern of translation through mTORc1-eIF4F-YY1.** The results so far described do not fully explain the protection that

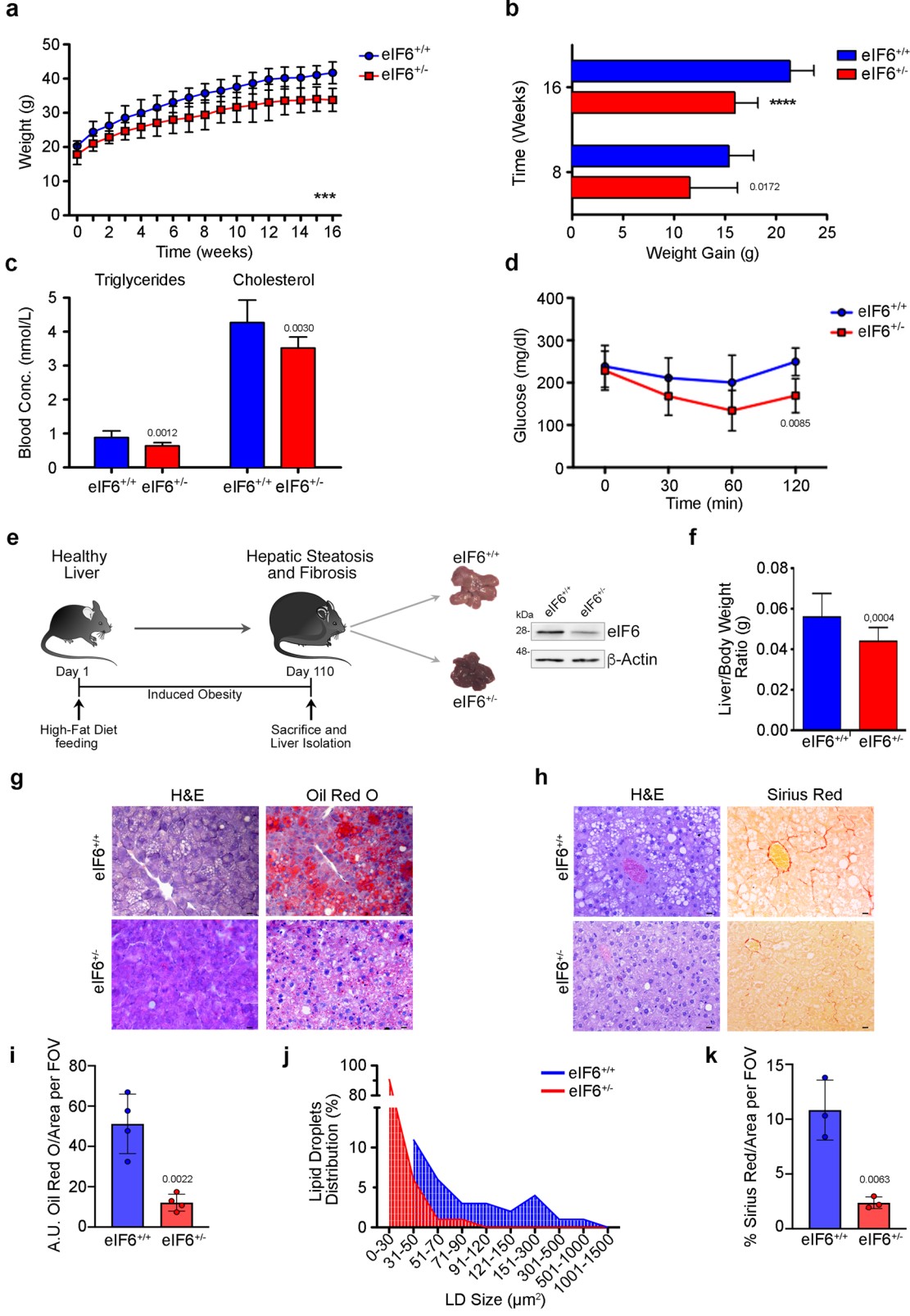

eIF6 depletion exerts also upon exogenous palmitate administration. Indeed, an additional feature of the eIF6 depletion-driven gene signature was the increase in the mitochondrial pathways (Fig. 4; Fig. 6a). Next, we isolated primary hepatocytes from obese eIF6$^{+/+}$ and eIF6$^{+/-}$ mice (Fig. 6b, c). As expected, we found less FASN (Fig. 6c) and reduced intracellular lipid droplets accumulation in eIF6$^{+/-}$ primary hepatocytes (Fig. 6d), compared with

eIF6$^{+/+}$. To further address mitochondrial functionality, we analyzed the fluorescence intensity of the mitochondrial marker Tomm20 (translocase of the mitochondrial outer membrane 20). Tomm20 staining was higher in eIF6$^{+/-}$ cells (Fig. 6e and Supplementary Fig. 5a). The same results were obtained using another marker, Aif[30] (Supplementary Fig. 5b). No nuclear staining of Aif indicating cellular apoptotic events was observed

**Fig. 3 eIF6 haploinsufficiency protects from hepatic steatosis and fibrosis and preserves insulin sensitivity. a** Growth curves of eIF6[+/+] and eIF6[+/−] mice fed with high-fat diet, HFD ($n = 12$ for eIF6[+/+] and $n = 15$ for eIF6[+/−]). Data are expressed as means ± SD. Two-way ANOVA. ***$p$ value ≤ 0.001. **b** Body weight gain is reduced in eIF6[+/−] mice compared with wt ones at the indicated time points ($n = 12$ for eIF6[+/+] and $n = 14$ for eIF6[+/−]). Two-tailed $t$ test. Data are represented as means ± SD. **c** Blood triglycerides and cholesterol levels are reduced in eIF6[+/−] mice at the end of HFD regimen ($n = 11$ for genotype). Two-tailed $t$ test. Data are represented as means ± SD. **d** Glucose levels in the blood: insulin tolerance test shows that eIF6[+/−] mice respond better to insulin administration at 120 min ($n = 5$ for genotype). Two-tailed $t$ test. Data are represented as means ± SD. **e** Outline of the experiment. Representative images of eIF6[+/+] and eIF6[+/−] livers are shown. Right: representative western blotting analysis of eIF6 levels in HFD-fed eIF6[+/+] and eIF6[+/−] mice. **f** Liver to body weight ratio shows that eIF6[+/−] livers are less enlarged compared with wild-type ones at the end of HFD feeding. Two-tailed $t$ test. Data are represented as means ± SD. ($n = 18$ for eIF6[+/+] and $n = 19$ for eIF6[+/−]). **g** Representative images of liver sections: staining with Oil Red O reveals reduced lipid droplets accumulation in eIF6[+/−] mice compared to wild-type ones. Scale bar = 20 μm. **h** Collagen fibers stained with Sirius Red: liver fibrosis is reduced. Scale bar = 20 μm. **i** Quantification of the area stained by Oil Red O. Data are expressed as a percentage of eIF6[+/+] controls. Hepatic lipid content is decreased in eIF6[+/−] mice. ($n = 4$ for genotype). Two-tailed $t$ test. **j** Representative frequency distribution of lipid droplets (LDs) size. Larger LDs were detected in eIF6[+/+] livers compared with eIF6[+/−] ones. **k** Quantification of area stained by Sirius Red. Hepatic fibrosis is decreased in eIF6[+/−] mice. ($n = 3$ for genotype). Data are provided as a Source Data File.

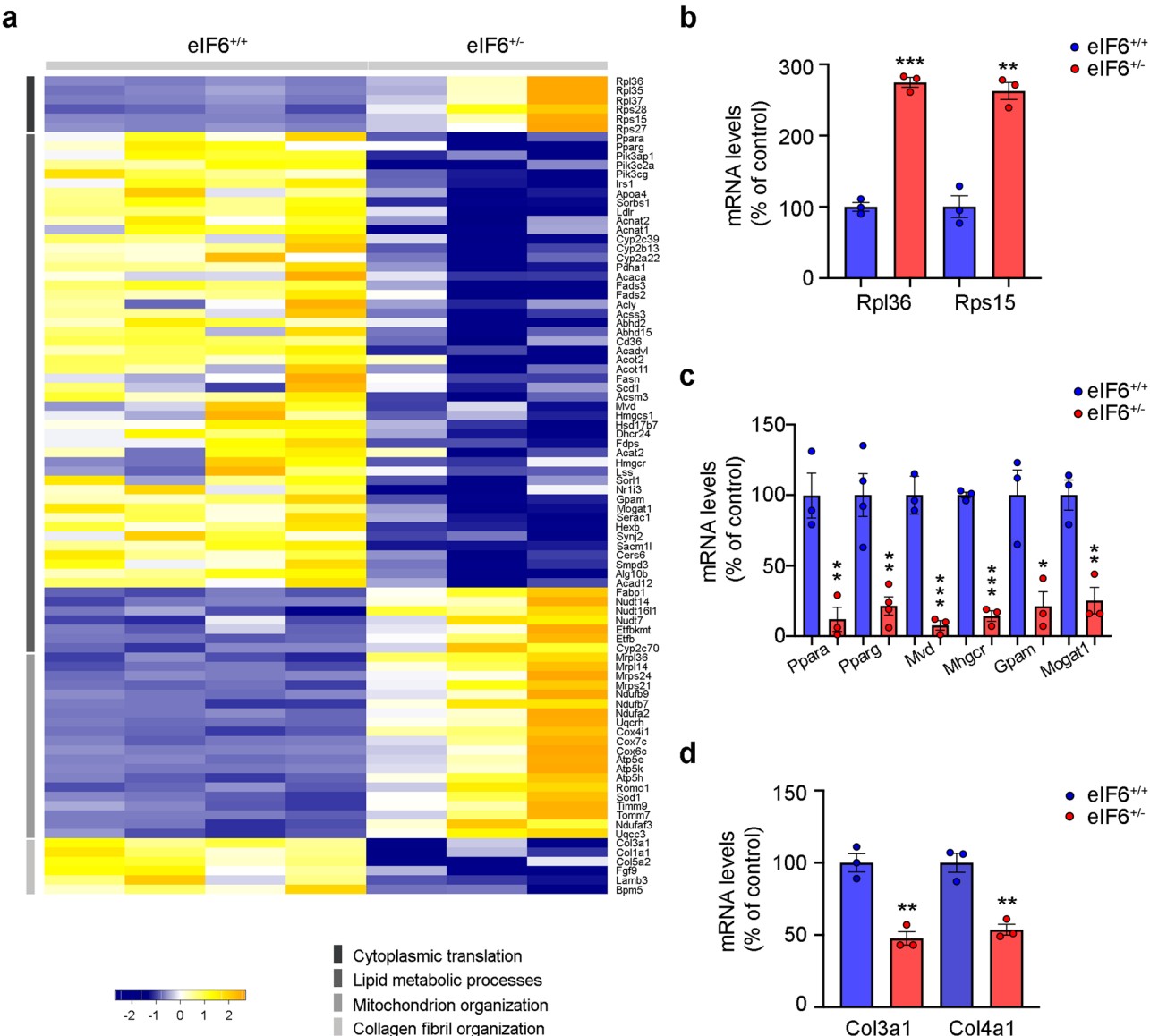

**Fig. 4 eIF6 haploinsufficiency affects hepatic lipogenesis. a** Heatmap of differentially expressed genes in eIF6[+/−] versus eIF6[+/+] livers: each column represents a single liver sample. Read count differences are visualized as $Z$-scores. Genes are clustered in four indicated biological processes. **b–d** Real-Time PCR analysis of selected targets validates RNA-Seq data ($n = 3$ for genotype). Data are represented as means ± SD. Two-tailed $t$ test. **$p$ value ≤ 0.01, ***$p$ value ≤ 0.001. Data are provided as a Source Data File.

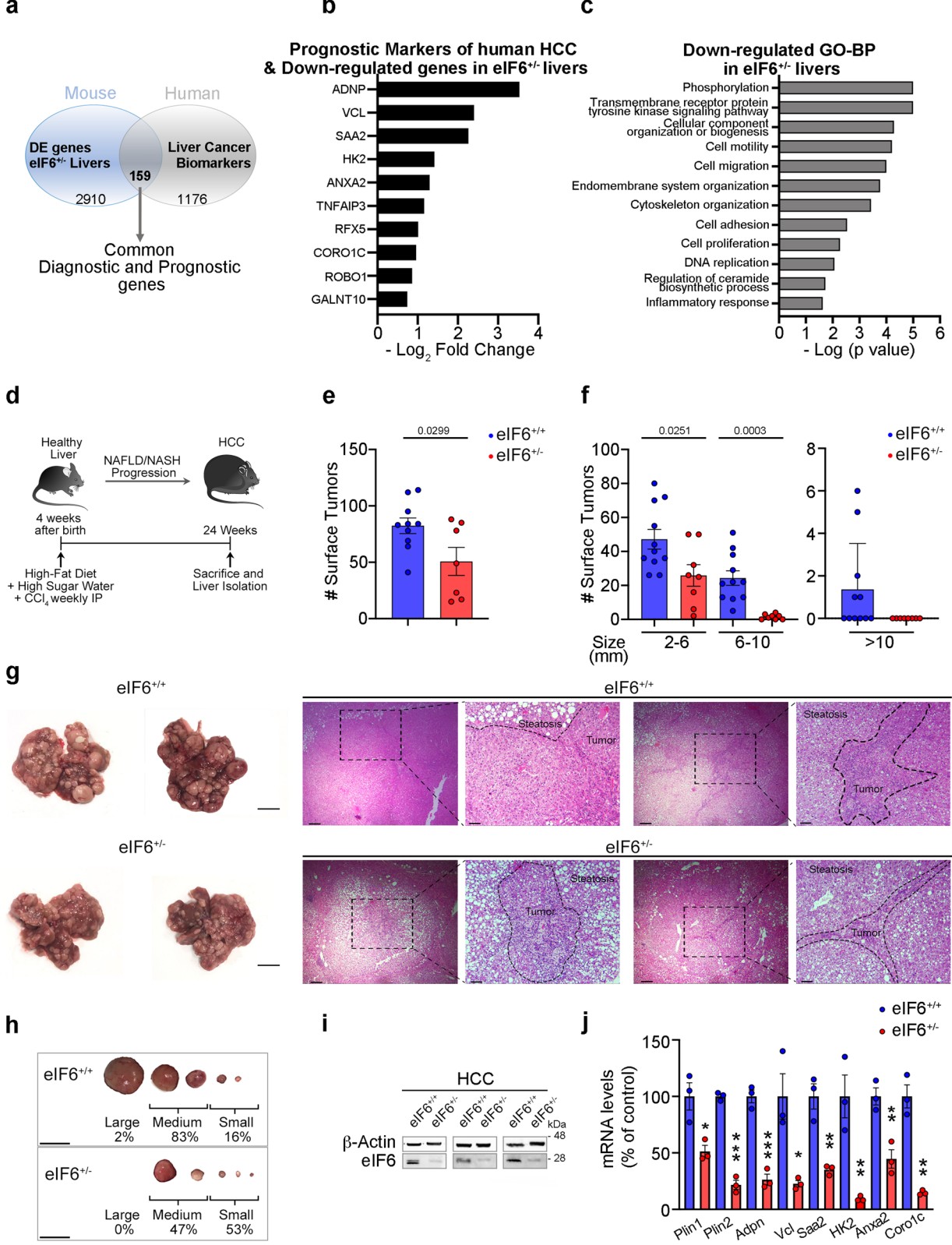

(Supplementary Fig. 5c). After the HFD regimen, intracellular ATP levels were higher in eIF6$^{+/-}$ hepatocytes (Fig. 6f), but not lactate production (Fig. 6g), compared with wild-type counterparts. These results stress the fact that eIF6 long-term depletion, during HFD, leads to lower lipid accumulation in primary hepatocytes, preserving mitochondrial integrity and without increasing lactate production, i.e., an evident anti-Warburg[5] effect.

The mitochondrial integrity driven by eIF6 deficiency was puzzling since no transcription factors were found directly regulated by eIF6 in the mitochondrial pathways. To explain the effect of eIF6 depletion on mitochondria, we performed a computational Functional Network Analysis (FGNet) on upregulated mitochondrial targets. We confirmed a functional network interaction between eIF6 and mitochondrial genes

**Fig. 5 eIF6 is required for NAFLD/NASH rapid progression into hepatocellular carcinoma. a** Comparison between prognostic and diagnostic markers of human hepatocellular carcinoma (HCC) and DE genes derived from RNA-Seq data of eIF6$^{+/−}$ livers: 159 common genes were found. **b** Representative downregulated genes in eIF6$^{+/−}$ livers identified as prognostic markers of human HCC. The histogram indicates the −Log$_2$ fold change. **c** Gene Ontology (GO-BP) analysis identifies the most significantly downregulated pro-tumoral pathways in eIF6$^{+/−}$ livers. **d** Schematic overview of NAFLD/NASH/HCC progression mouse model. **e, f** Quantification of total surface tumors number (left) and size distribution (right) in 24 weeks eIF6$^{+/+}$ ($n = 11$) and eIF6$^{+/−}$ ($n = 8$) livers. Two-tailed $t$ test. **g** Gross appearance of livers and surface neoplastic nodules (left). Scale bars = 1 cm. H&E staining of liver sections of HFD/CCl4 mice at 24 weeks. Dashed black lines outline tumor areas surrounded by steatotic tissue. Scale bars = 500 μm (×4 magnification) and = 50 μm (×10 magnification). **h** Nodules size (as a percentage). Scale bars = 1 cm. **i** Representative western blotting for eIF6 on livers of HFD/CCl4 mice. **j** QT-PCR analysis of selected genes involved in lipid storage or associated with prognosis of HCC ($n = 3$ per genotype). Data are represented as means ± SD. Two-tailed $t$ test. *$p$ value ≤ 0.05, **$p$ value ≤ 0.01, ***$p$ value ≤ 0.001. *NAFLD* nonalcoholic fatty liver disease, *NASH* nonalcoholic steatohepatitis. Data are provided as a Source Data File.

(Fig. 7a). We detected interactions with genes encoding for the mitochondrial electron transport chain (ETC) complex I, III, IV, and V (Fig. 7b). After validation of representative genes of this network (Fig. 7c), we hypothesized that the mitochondrial factors that increased at the mRNA level in eIF6$^{+/−}$ mice were regulated by a common transcription factor. Thus, we analyzed the 5′ region of these mitochondrial genes, in order to identify *cis*-acting regulatory elements which may control their transcription. In short, we identified common motifs recognized by the transcription factor Yin-Yang 1, YY1 (Fig. 7d), which is important for mitochondrial biogenesis[31]. Next, we tested the expression of YY1 in HFD eIF6$^{+/−}$ livers. We found by western blotting that YY1 protein levels were higher in age-matched HFD eIF6$^{+/−}$ livers (Fig. 7e; f). This result was associated with an increase of YY1 mRNA on the heavy polysomes of eIF6$^{+/−}$ livers (Fig. 7g). This last observation could be explained if translational initiation relied on the maintenance of a signaling cascade alternative to eIF6, possibly through mTORc1-eIF4F[18,32,33].

We, therefore, addressed whether (a) the translation rate and the mTORc1 pathway are protected by eIF6 depletion in the chronic setup of HFD, (b) YY1 mRNA contains 5′-UTR elements that confer sensitivity to mTORc1 inhibition, and (c) YY1 affects FAO. We found that (a) the global translational rate, measured by puromycin incorporation in vivo, is progressively reduced during the worsening of hepatic steatosis in eIF6$^{+/+}$ mice (Supplementary Fig. 6a–c), and that at the end of the HFD regimen, eIF6$^{+/−}$ livers had higher levels of puromycin incorporation (Supplementary Fig. 6a–c), and slightly higher polysome/80 S ratio, compared with wt mice. In addition, two major downstream mTORC1 kinase substrates, rpS6 and 4E-BP1, were more phosphorylated in eIF6$^{+/−}$ livers (Supplementary Fig. 6e). Taken together, these results unequivocally demonstrate that the mTORc1 translation pathway is spared by eIF6 depletion, in the chronic setup of HFD.

Next, (b) we identified the 5′-UTR of YY1 and cloned it in front of a luciferase reporter (Supplementary Fig. 6f, g). By luciferase assays, we found that the translation of a reporter gene with the 5′-UTR of YY1 depended on mTORC1 activity, as it was strongly repressed by rapamycin (Supplementary Fig. 6h). Finally, (c) we addressed if YY1 affects FAO, in the specific context of eIF6 partial depletion. In AML12 hepatocytes, eIF6 silencing increases FAO, measured by [14]C-palmitate oxidation (Fig. 7h, i), at levels similar to the ones obtained by the AMPK activator, AICAR (5-aminoimidazole-4-carboxamide ribonucleoside). Silencing of YY1 in eIF6-depleted cells reverses the phenotype and leads to a reduction of fatty-acid β-oxidation (Fig. 7h, i). Taken together, the data suggest that inhibition of eIF6 activity directly inhibits the translation of lipogenic transcription factors and spares a translation pathway that maintains mitochondrial functionality via YY1.

We attempted to see if evidence of deregulated YY1 pathway existed in human data sets. Metadata analysis indicated that during the progression of NAFLD mRNA levels of YY1 decrease (Fig. 7j). Furthermore, YY1 mRNA levels and its upregulated

targets in conditions of eIF6 depletion individually showed a consistent trend with benign prognosis in primary liver cancer (Supplementary Fig. 7). In conclusion, mitochondrial fitness is maintained in conditions of chronic eIF6 depletion/HFD due to an alternative translationally controlled pathway.

**Pharmacological eIF6 targeting elicits an anti-lipogenic effect.** eIF6 is a rate-limiting factor for the translational regulation of transcription factors necessary for de novo lipogenesis, such as C/EBPβ. This effect occurs at the molecular level by re-initiation (the model in Fig. 8a). Recently, we demonstrated that eIF6 activity is modulated by newly identified compounds that inhibit eIF6 binding to the 60 S[34]. This prompted us to test whether the pharmacological targeting of eIF6 reduces lipogenesis and lipid accumulation in an in vitro model of steatosis and C/EBPβ polysomal association. Upon palmitate addition, AML12 cells undergo cellular lipid droplet accumulation. We found that lipid accumulation is reduced by eIFsixty-1 and eIFsixty-6, two inhibitors of the binding of eIF6 to 60 S subunits, as compared with negative controls that do not block the binding of eIF6 to 60S (Fig. 8b, c). In agreement, both eIFsixty-1 and eIFsixty-6 decrease the polysomal association of the lipogenic transcription factor C/EBPβ (Fig. 8d and Supplementary Fig. 8a). In contrast, mTORc kinase activity and translation of YY1 mRNA are maintained upon eIFsixty-1 and administrations (Fig. 8d and Supplementary Fig. 8b). eIFsixty-1 and eIFsixty-6 maintain 4E-BP1 phosphorylation and YY1 expression at the protein level and lead to a drop of all C/EBPβ translation products (Supplementary Fig. 8c). Overall, these findings provide evidence that eIF-sixty inhibitors are able, at least to some extent, in vitro, to recapitulate the mechanistic and anti-steatotic effects of eIF6 depletion.

## Discussion

The background of our study was the discovery of translational control of lipid synthesis driven by eIF6[5,14], combined with the fact that eIF6 reduction impairs tumor progression[13]. We hypothesized that the combination of the effects of eIF6 on lipid synthesis and on malignancy is relevant in the liver, specifically in the progression from NAFLD to HCC. Our data confirmed the expectations. We will discuss our findings in the context of recent evidence, suggesting that protein synthesis is an important determinant for metabolism[5].

Nutrient-sensing pathways optimize energy consumption during starving periods and amplify energy storage in conditions of a burst of nutrients. At the same time, nutrient availability has a stimulatory effect on protein synthesis raising the question of whether the regulation of translation may also impact energy storage, such as lipid accumulation. Indeed, we found that the postprandial protein synthesis program in the liver leads to the specific translation of mRNAs encoding transcription factors that

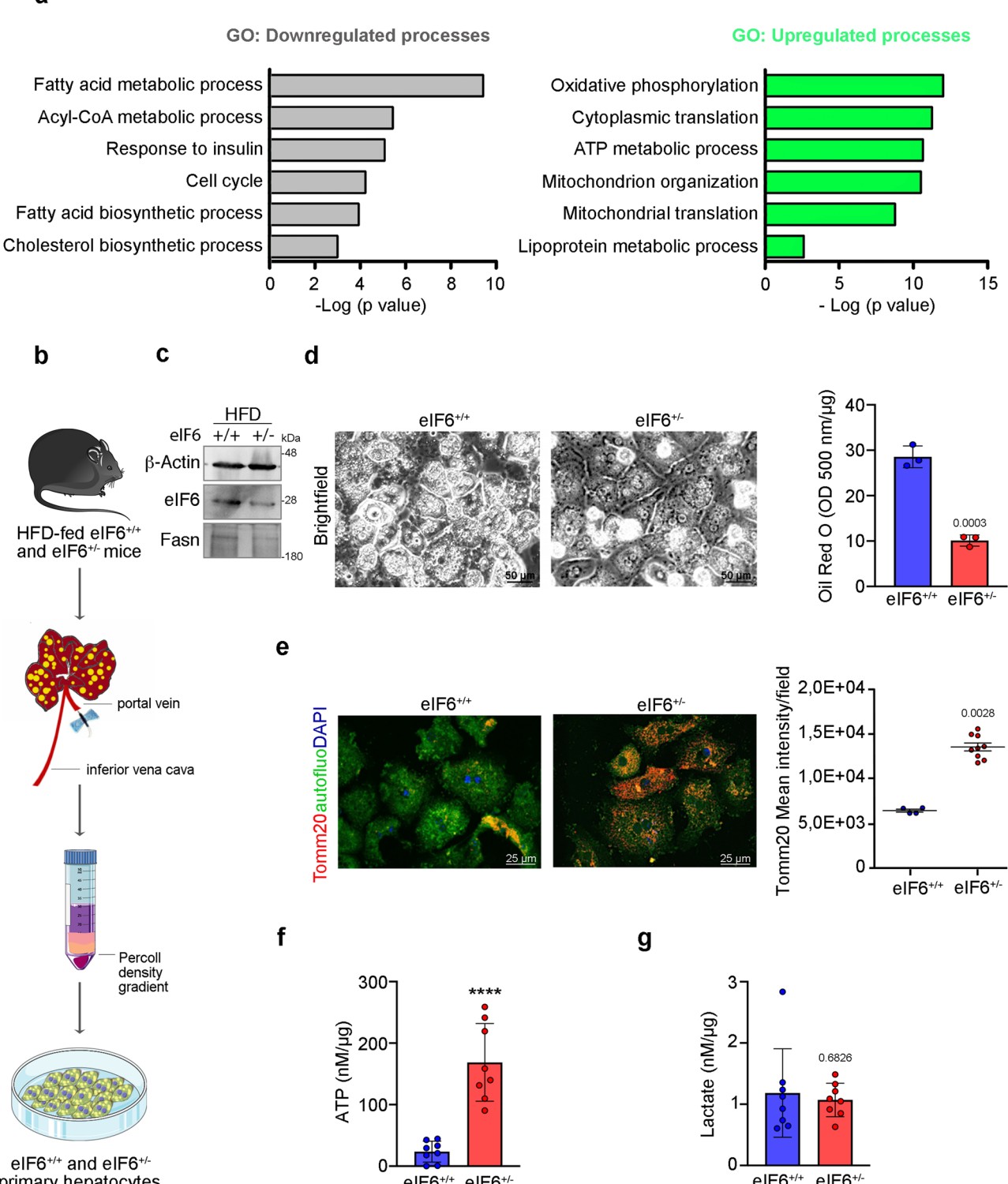

**Fig. 6 eIF6 depletion preserves mitochondrial bioenergetic capacity. a** Gene ontology (GO) analysis identifies upregulated mitochondrial pathways in eIF6$^{+/-}$ livers. **b** Scheme showing the primary hepatocytes isolation outline. **c** Representative western blotting of eIF6 and FASN in primary hepatocytes, isolated from HFD-fed mice. **d** Left: bright-field images of primary hepatocytes isolated from eIF6$^{+/-}$ and eIF6$^{+/+}$ obese mice. eIF6$^{+/+}$ hepatocytes have an increased accumulation of lipid droplets. Scale bars are indicated. Right: quantification of Oil Red O staining in primary hepatocytes. Data are represented as means ± SD, $n = 3$ for genotype. Two-tailed $t$ test. **e** Left: immunofluorescence of primary hepatocytes stained for Tomm20 (red) and DAPI (blue). Autofluorescence (green) was used to visualize cells. Scale bars are indicated. Right: quantification of Tomm20 fluorescence intensity levels shows that they are increased in eIF6$^{+/-}$ hepatocytes. Mann–Whitney test. **f, g** eIF6$^{+/-}$ primary hepatocytes have increased ATP levels (**f**) but similar lactate secretion (**g**). Two-tailed $t$ test. Data are represented as means ± SD of independent samples. Data are provided as a Source Data File.

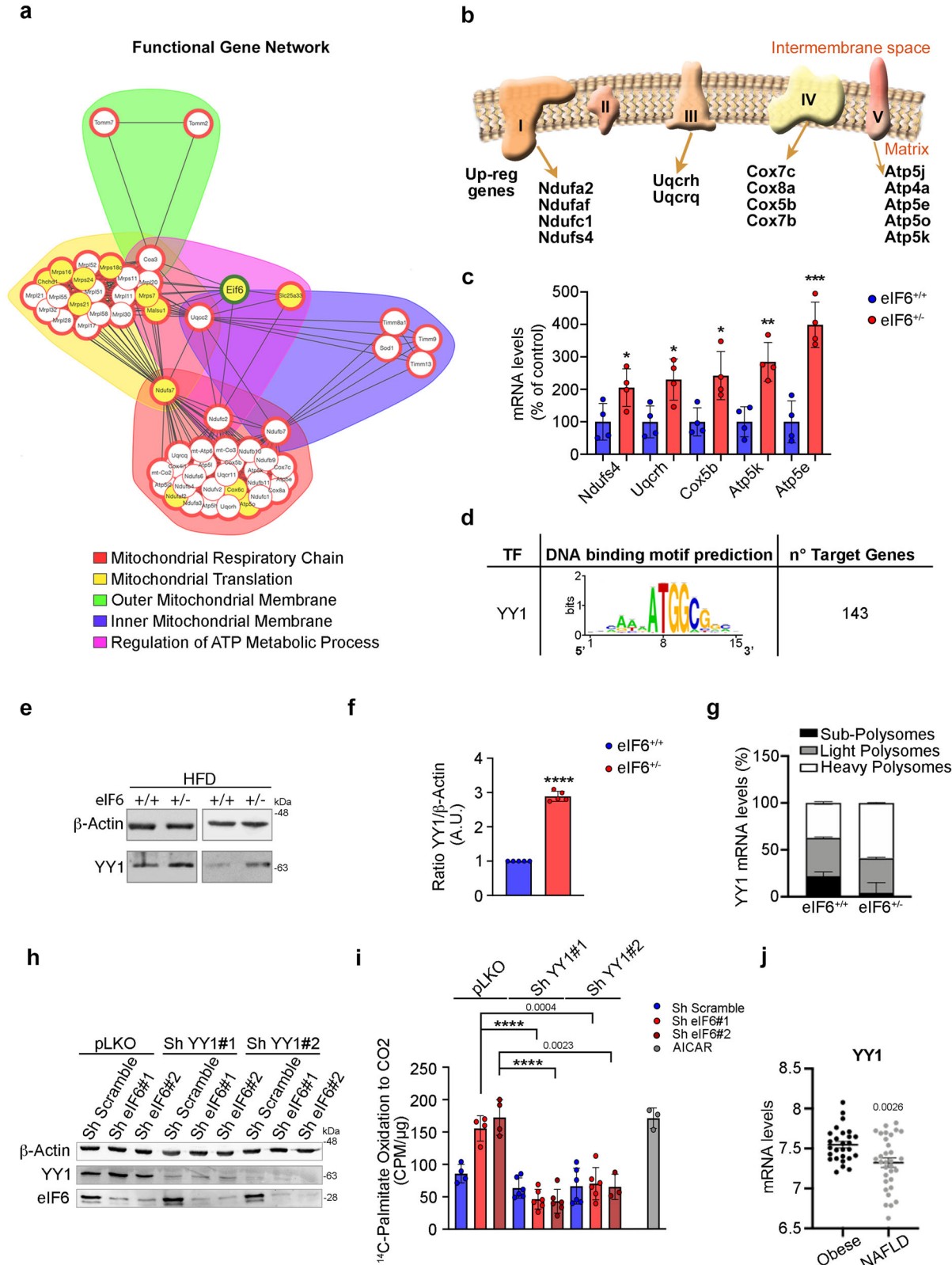

sustain a lipogenic program[5]. In evolutionary terms, this response may sustain for a longer time, and more effectively, energy storage through the preferential accumulation of fat. We define this process metabolic learning, similar to long-term potentiation in the nervous system, where translation becomes necessary for the transition from short-term memory to long-term memory[35]. If metabolic learning exists, then the inhibition of the translational

response driven by high nutrients had to impact the NAFLD phenotype, as it should overturn the "accumulation" scope of the nutrient sensory pathway. In addition, given the fact that nutrient sensory pathways are always altered in tumor cells[33], the impact of blocking the translational increase driven by high glucose and insulin was expected to affect also the progression to HCC. This is indeed what we observed after the inhibition of eIF6, a reduction

**Fig. 7 Mitochondrial target genes regulation is mediated by the spared mTORC1-YY1 axis. a** Functional Gene Network represents the mitochondrial genes correlated to eIF6 expression. **b** Simplified scheme of upregulated ETC genes identified by RNA-Seq analysis. **c** Real-time PCR of selected mitochondrial targets ($n = 4$ for genotype). Two-tailed $t$ test. *$p$ value $\leq 0.05$, **$p$ value $\leq 0.01$, ***$p$ value $\leq 0.001$. **d** YY1-binding motif predicted in mitochondrial genes associated with eIF6 expression. **e** Representative western blotting analysis shows that YY1 protein expression level is increased in eIF6$^{+/-}$ livers compared with wt ones. **f** Densitometric analysis of YY1 expression normalized on β-actin. Data are expressed as an arbitrary unit ($n = 5$ per genotype). Data are represented as means ± SD. Two-tailed $t$ test. **g** Real-time PCR of YY1 mRNA on liver polysome fractions reveals that its association with heavy polysomes (white box) is increased in eIF6$^{+/-}$ samples compared with wt ($n = 3$). Stacked bar charts represent the quantification of selected mRNA levels in heavy, light, and subpolysomes. **h, i** FAO (fatty-acid oxidation) rates measurement in all indicated conditions. Western blots of analyzed samples (**h**) and measured counts (**i**). YY1 depletion reduces $^{14}$C-palmitate Oxidation to $CO_2$. Data are indicated as percentage of control (pLKO-ShScramble transduced cells) and represented as means ± SD ($n = 4$). Two-tailed $T$ test. **j** NAFLD patients show a reduced expression of YY1 mRNA. Data are presented as mean ± SEM. Two-tailed $t$ test. *HFD* high-fat diet. Data are provided as a Source Data File.

in the lipid accumulation program, accompanied by a reduced progression of HCC.

Whereas chronic eIF6 depletion reduced lipogenesis, we found that mitochondrial fitness was spared by the reduction of eIF6 activity. Interestingly, this finding can be at least partly explained by the differential regulation of translation by nutrient-regulated signaling cascades[18]. Growth factor tyrosine kinase receptors can simultaneously stimulate the PI3K-mTORC1, RAS-MAPK, and PLCγ-PKC pathways. eIF6 is activated independently from the PI3K-mTORC1 pathway[36]. In conditions of eIF6 depletion, the signaling driven by mTORC1 to the translational machinery is therefore maintained. Morita et al.[37] reported that the mTORC1 pathway translationally controls mitochondrial activity. Consistently, we found that YY1, which is important for mitochondrial biogenesis[38], and is necessary to sustain the mitochondrial oxidative capability driven by mTOR[39], is translationally regulated by mTORC1 and its downstream effector eIF4F. Thus, eIF6 depletion contributes to the protection from lipid accumulation both directly, reducing the fatty-acid synthesis and, indirectly, preserving FAO.

eIF6$^{+/-}$ mice have reduced postprandial translation[14]. At first, we found puzzling that after HFD, the liver of eIF6$^{+/-}$ mice translated more than the liver of wt mice. We, therefore, reconstructed the modulation of translation during disease progression. Consistently with previous work[14], we found that at early HFD the wt liver translates more because eIF6 is rate-limiting. Higher levels of eIF6, however, exacerbate the lipogenic action of the HFD. With increasing damage, the wt liver reduces translation, whereas the eIF6-depleted liver is healthier and maintains protein synthesis. These findings are in agreement with the model that eIF6-mediated increased translation exacerbates HFD damage, and then HFD damage reduces mTORC1-dependent translational capability. A mutually not exclusive hypothesis is that insulin sensitivity is protected in the liver of eIF6$^{+/-}$ mice, and leads to optimal activation of the mTORC1 pathway.

We will briefly discuss the case of PPARγ. Globally, the targets inhibited by eIF6 depletion are thought to speed up the progression of NAFLD. Among them we found, ACC1, FASN, FXR[1], as well as fibrosis markers produced by stellate cells[1,40]. Notably, in conditions of eIF6 depletion, the levels of PPARγ drop. This result goes against the general trend because PPARγ agonists have been proposed to therapeutically improve NAFLD[1], in line with genetic evidence that indicates that its ablation worsens the effects of an HFD[41]. Mechanistically, PPARγ downregulation can be easily explained by the effects of eIF6 on C/EBPβ, and is consistent with the fact that deleting the uORF of C/EBPβ results in improved lipid metabolism owing to LIP ablation[42]. This said we hypothesize that the extensive transcriptional changes induced by eIF6 depletion are sufficient to induce a beneficial effect in the presence of PPARγ downregulation. Alternatively, PPARγ may be regulated at the posttranscriptional level. At the pharmacological level, it will be interesting to study whether PPARγ stimulation further attenuates the NAFLD phenotype inhibited by eIF6 depletion. Our data,

concluding, suggest that interfering with the early translational events linked to food consumption, overcomes the need for PPARγ stimulation, which possibly is an adaptive event to HFD.

We provide proof-of-concept that eIF6 modulators[34] can, in vitro, modulate in an expected way the translation of eIF6 targets, protecting the expression of YY1. The long-term effects of these compounds, including their specificity, must be thoroughly addressed in future studies. The development of specific eIF6 inhibitors that may be used also in vivo is greatly needed.

In conclusion, our data on eIF6 show that a feed-forward translational mechanism exacerbates the negative effects of lipid accumulation, as recently suggested also for eIF4E[43]. We hypothesize that translational regulation of lipid metabolism can be a therapeutic target in the NAFLD-HCC progression.

## Methods

**Human data sets.** The following data sets were used. The microarray cohort contains 32 NAFLD patients and 27 healthy obese patients[15]. The RMA algorithm was employed on the raw data for normalization of the data sets and expression data were retrieved. Statistical analysis was performed by a two-tailed $t$ test. The second cohort consists of 125 severely obese individuals who underwent percutaneous liver biopsy during bariatric surgery[17]. Individuals with increased alcohol intake (>30/20 g/day in M/F), viral and autoimmune hepatitis, or other causes of liver disease were excluded. For histological studies, we used liver sections derived from three different types of patients, classified as mild (non-severely-obese patients), moderate (simple steatosis), and severe (steatosis with progressive fibrosis and NASH), based on a scoring system[44] defined by anatomic pathologists with expertize in NAFLD, NASH, and HCC ($n = 12$). Informed consent was obtained from each patient and the study protocol was approved by the Ethical Committee of the Fondazione IRCCS Ca' Granda and conformed to the ethical guidelines of the 1975 Declaration of Helsinki. eIF6$^{+/-}$ DE genes were also matched to a list of 25 genes that connects dyslipidemia to HCC[27] (Supplementary Table 2). Then, we crossed eIF6$^{+/-}$ DE genes with a list of liver cancer biomarkers found in CancerLivER (Liver Cancer Expression Resource)[28] (Supplementary Table 3).

**Experimental procedures on eIF6 transgenic mouse model**

*Models.* eIF6$^{+/+}$ and eIF6$^{+/-}$ transgenic mice were generated, backcrossed to C57BL/6 N strain ($N$ generation $\geq 29$), and genotyped[45]. For this study, a cohort of age-matched mice ($n = 12$ for eIF6$^{+/+}$ and $n = 15$ for eIF6$^{+/-}$) were fed with HFD (Research Diet D12451, containing 45 Kcal% fat). HFD feeding started after weaning (4 weeks old mice) and was maintained for 16 weeks. To monitor food intake, a cohort of eIF6$^{+/+}$ and eIF6$^{+/-}$ mice were individually housed and acclimatized for 1 week before the study. Energy intake was measured every 3 days for 30 days. Upon HFD regimen, both eIF6$^{+/+}$ and eIF6$^{+/-}$ mice were weighted as indicated in the manuscript. At the end of the 16-week HFD-feeding regimen, blood from eIF6 tg mice was collected by cardiac puncture, and serum was loaded on test-strips to measure cholesterol, triglycerides, and glucose levels, using the MULTICARE "IN" analyzer. ITT was performed. Comparative analyses were performed in blind. At the end of the HFD-feeding regimen, all experimental mice were killed. Mouse Model of NAFLD/NASH rapid progression to HCC: 4 weeks eIF6$^{+/+}$ and eIF6$^{+/-}$ male mice were fed with HFD diet (Research Diet D12451, containing 45 Kcal% fat) and a high sugar water solution (23.1 g/L D-fructose, Sigma-Aldrich, F0127; 18.9 g/L D-glucose, Sigma-Aldrich, G8270)[29]. This diet regimen started simultaneously with weekly i.p. injection of CCl4 (0.2 μl/g of BW; Sigma-Aldrich, 289116). Mice were killed after 24 weeks of HFD and CCl4-treatment and liver samples were collected and processed for morphological, histological, and gene expression analysis.

*In vivo puromycin labeling.* The hepatic translational rate was measured by puromycin labeling. Randomized HFD-fed male mice at 8, 12, and 16 weeks from the

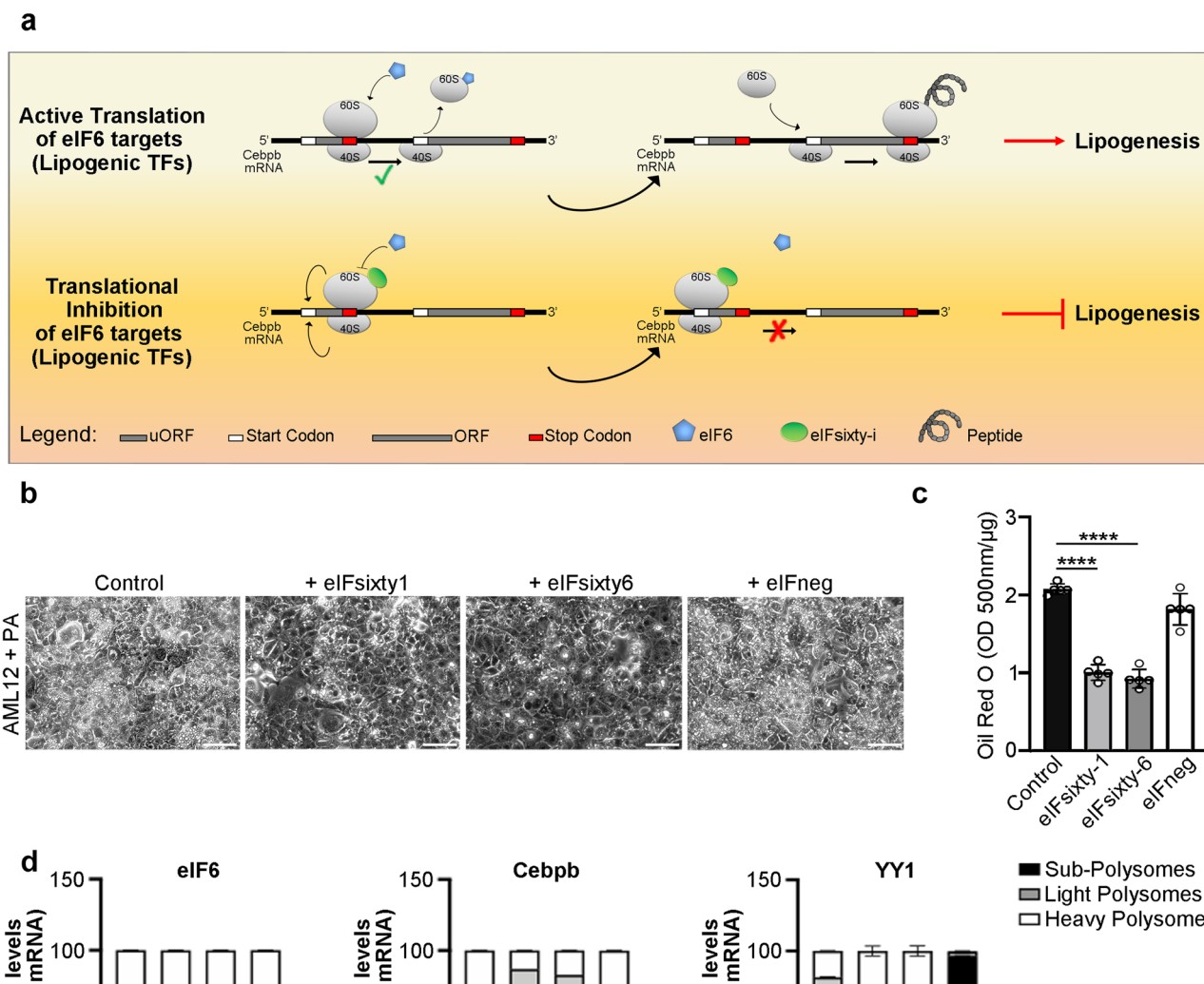

**Fig. 8 Pharmacological targeting of eIF6 activity reduces steatosis in murine hepatocytes. a** Model of the translational regulation of lipogenic TFs driven by eIF6. eIF6 regulates the translation of C/EBPβ mRNA, which contains uORF in its 5′UTR, through re-initiation. eIF6 has an anti-association property: it binds the 60 S ribosomal subunit at the stop codon of the uORF, leading to 60 S release from the 40 S. 40 S can then resume scanning to the main ORF, which encodes for the transcription factor (TOP). eIFsixty compounds prevent eIF6 binding to the 60 S at the stop codon of the uORF. In this case, the inactive 80 S complex is stalled and the translation of the downstream ORF is inhibited (BOTTOM). **b** Bright-field images of AML12 treated with 2 mM palmitate and with eIFsixty-i compounds. Cells treated with palmitate have increased accumulation of lipid droplets, which is reduced upon eIFsixty-1 and eIFsixty-6 inhibitors treatment. eIFneg is a negative control, an inactive compound of a similar structure. Scale bar = 50 μm. **c** Quantification of Oil Red O content in palmitate-treated AML12 cells. $N = 5$/condition. Data are represented as means ± SD. Two-tailed $t$ test. **d** Quantification of mRNA levels in heavy, light, and subpolysomes of AML12 cells after administration of palmitate and eIFsixty-i compounds. eIFneg was used as a negative control. Only eIFsixty-1 and eIFsixty-6 induce a shift from the polysomal to the subpolysomal peak of eIF6 target C/EBPβ mRNA. The same treatment shows the opposite effect on YY1 mRNA. Real-time PCR analysis was performed on $n = 3$ independent experiments. Stacked bar charts represent the quantification of selected mRNA levels in heavy, light, and subpolysomes. Data are provided as a Source Data File.

start of the HFD regimen were fasted for 16 h and then refed. In all, 1 h after refeeding, mice were injected intraperitoneally with 50 mg/kg body weight puromycin. In all, 3 h after IP injection, mice were killed and livers samples were isolated and processed for histological analysis and for puromycin detection by western blotting using anti-puromycin antibodies (Sigma-Aldrich, #MABE343). The quantitation of puromycin staining was performed measuring DAB intensity through the Image J software. eIF6$^{+/+}$ and eIF6$^{+/-}$ mice not injected with puromycin were used as negative controls.

All mice were maintained under specific and opportunistic pathogen-free conditions and all experiments involving animals were performed in accordance with the Ethical Committee of San Raffaele by experimental protocols approved by national regulators (IACUC no. 688). Mandatory rules for 3Rs and euthanasia were followed.

**RNA isolation and Illumina sequencing**. Approximately 5 mg of liver tissue from standard or HFD mice were homogenized. RNA was purified using mirVana™ isolation kit (ThermoFisher) according to the manufacturer protocols and as

previously described[46,47]. RNA quality was assessed using the Agilent RNA Pico 6000 Kit and BioAnalyzer (Agilent Technologies). Random primed cDNA libraries for Illumina sequencing were constructed from 100 ng of total RNA. A paired-end (2 × 125) run was performed on a Genome Sequencer Illumina HiSeq 2500[17] (GATC Biotech). Sequence depth was at 30 million reads.

**RNA-seq data analysis.** Four biological replicates for HFD-fed eIF6[+/+] and three biological replicates for HFD-fed eIF6[+/−] mice were analyzed. All mice were age- and sex-matched. The pipeline of analysis was as previously described[47]. In brief, raw reads were checked for quality by FastQC software and filtered to remove low-quality calls by Trimmomatic. Processed reads were then aligned to the mouse genome (Ensembl version 87) with STAR software. HTSeq-count algorithm with default parameters (gene annotation release 83 from Ensembl) was employed to produce gene counts for each sample. To estimate differential expression was used DESeq2 package. The differential expression analysis by the DeSeq2 algorithm was performed on HFD-fed eIF6[+/−] versus HFD-fed eIF6[+/+] livers (seven samples overall). Reads counts were normalized by calculating a size factor, as implemented in DESeq2. An Independent filtering procedure was then applied, setting the threshold to the 62 percentile. In all, 25,695 genes were therefore tested for differential expression. Significantly modulated genes were selected by considering a false discovery rate lower than 10%. Subsequent computational analysis was performed comparing eIF6 genotypes as reported in the manuscript. Regularized logarithmic (rLog) transformed values were used for heatmap representation of gene expression profiles of the HFD-fed eIF6[+/−] versus HFD-fed eIF6[+/+] comparison. Analyses were performed in R version 3.3.1. The Gene Ontology enrichment analysis was performed using the topGO R Bioconductor package (version topGO_2.24.0) as described[46]. The annFUN.db function was used to extract the gene-to-GO mappings from the genome-wide annotation library org. Mm.eg.db for *Mus Musculus*. Functional Gene Networks (FGNet), an R/Bioconductor package[48] was used to identify the functional association network of the DEGs (differentially expressed genes). Ball-nodes represent genes and links represent the predictive functional association between genes based on biological annotations. For Cis-regulatory motif analysis, the RSEM tool was used to quantify different transcript isoforms abundance of DEGs. Next, Weeder2.0 was used for the de novo motif (transcription factor binding sites) discovery. To validate enriched transcription factor motifs and their optimal sets of direct targets., the obtained results were matched with JASPAR CORE database[49,50] and iRegulon plugin. Only the TFs shared by all analyses were considered.

**mRNA extraction and quantitative RT-PCR.** Total RNA from livers of transgenic mice was extracted with TRIzol reagent (Invitrogen). mRNAs collected from fractions of sucrose gradient during polysome profiling analysis were divided into subpolysomal (from the top of the gradient to the 80 S peak), light polysomes, and heavy polysomes mRNAs[47]. Samples were incubated with proteinase K and 1% sodium dodecyl sulfate (SDS) for 1 h at 37° C. RNA was extracted by the phenol/chloroform/isoamyl alcohol method. After treatment of RNA with RQ1 RNase-free DNase (Promega), reverse transcription was performed according to SuperScript III First-Strand Synthesis kit instructions (Thermo Fisher Scientific). Complementary cDNA (100 ng) was amplified with the appropriate primers using StepOne Plus Real-Time PCR System (Thermo Fisher Scientific). For RNA-seq validation, both TaqMan and Syber green probes were used as indicated in the Supplementary Information file.

Target mRNAs quantification was performed by using the $\Delta\Delta C_t$ method with 18 s rRNA or β-actin mRNA as an internal standard. For the analysis of targets from polysome fractions, the data are quantitated as the percentage of expression in each fraction, normalized to the total amount of the target. Data points are means ± sd of three independent experiments.

**Histological staining and immunohistochemistry.** Human liver biopsies and murine samples were processed for histological analysis. Human samples were derived from non-severely obese patients with simple steatosis or progressive fibrosis and NASH, whereas murine liver samples were recovered at the end of the HFD-feeding experiment. In brief, all samples were stained with Hematoxylin and Eosin (Sigma-Aldrich) for morphological analysis[51]. Paraffin-embedded sections of human liver biopsies were treated with Tris-ethylenediaminetetraacetic acid (EDTA) buffer pH = 9 for 30' at 96° for the antigen retrieval and processed for immunohistochemical staining for Phospho-rpS6 (Phospho-S6 Ribosomal Protein (Ser235/236) Antibody #2211, 1:200, o/n) and eIF6 (Rabbit polyclonal anti-eIF6[52], 1:200, 0/n) using the Vectastain Elite ABC kit (DBA), according to the manufacturer's instructions[53]. Paraffin-embedded or OCT-embedded murine liver biopsies were subjected to Oil Red O staining (Sigma-Aldrich), Sirius Red staining (Sigma-Aldrich), according to the manufacturer's instructions. For quantification of Oil Red O areas, the images were subsequently analyzed using ImageJ. Change in the relative frequency distribution of lipid droplet size was measured clustering lipid droplets size in ascending intervals. DAB staining intensity of Phospho-rpS6 and eIF6 signals was normalized by the nucleus and quantified by ImageJ. For quantification of Sirius Red Area, red-stained collagen was selected and measured by ImageJ.

**Cell cultures.** We used 293 T (ATCC CRL-3216) and AML12 (ATCC CRL-2254) cells. Cells were tested monthly for mycoplasma and grown in Dulbecco's modified Eagle's medium (DMEM)/Ham's F12 (GIBCO) supplemented with 10% fetal bovine serum, a mixture of insulin, transferrin, and selenium (Invitrogen), 40 µg ml$^{-1}$ dexamethasone, and penicillin/streptomycin/glutamine solution (AML12) or fully supplemented as above but with DMEM (293 T). AML12 cells were treated for 72 h with palmitate 2 mM, and with eIF6ixty-i compounds at their IC50 concentration (1,4 µM for eIF6ixty-1; 1,1 µM for eIF6ixty-6). Oil Red O staining was performed. Colony transformation assay was performed on AML12 cells transduced with a retrovirus carrying DNp53 + oncogenic H-rasV12[12]. After 2 days these cells were also infected with lentiviral vectors carrying scramble shRNA as control or two eIF6-specific shRNAs, as previously described[12]. In all, 2–3 weeks after infections, transformed colonies were acquired with Bright-field microscopy as representative images and later stained with crystal violet for quantification. Scramble and eIF6 shRNAs-infected AML12 cells were further transduced with lentiviral vectors carrying pLKO shRNA as control or two YY1-specific shRNAs and FAO were evaluated. YY1 protein levels were detected by western blotting analysis (Santa Cruz Biotechnology, Cat#sc-7341). To perform rescue experiments, AML12 cells were transduced with lentiviral vectors carrying Ires-Neo construct as control or C/EBPβ-LIP construct, kindly provided by C.F. Calkhoven lab[26]. Overexpression of C/EBPβ isoforms was assessed by western blotting (Purified anti-C/EBP cat#606202, Biolegend) and Oil Red O staining was performed. EMSC was used for adipogenesis studies. Mesenchymal stem cells from outer ears of eIF6[+/+] and eIF6[+/−] mice were collected, as previously described[12]. Primary eIF6[+/−] EMSC were transduced with the lentiviral vector eIF6wt-pCCL to rescue eIF6 protein levels and GFP-pCCL, as control. The cells were later differentiated into adipocytes and stained with Oil Red O to measure lipid droplets accumulation, as previously described[14].

**Primary hepatocytes isolation from HFD-mice and Oil Red O quantitation.** Three-month-old mice were anesthetized by intraperitoneal injection of Avertin. Hepatocyte isolation was carried out essentially as previously described[14], except for the use of a 30% PERCOLL cushion (Amersham). The pellet of viable cells was suspended in DMEM-F12 and recovered at 5% $CO_2$, 37° C for 16 h before starting the experimental procedure. Matched littermates were analyzed in parallel. Hepatocytes were fixed in 10% buffered formalin and stained with Oil Red O for 10 min. The dye retained by the cells was eluted with isopropanol and quantified by measuring absorbance at 500 nm. Data were normalized to total cellular proteins (µg). All the analyses were performed at least three times.

**ATP content and lactate secretion.** Primary hepatocytes were lysed in 20 mM Tris-HCl (pH 7.5), 25 mM NaCl, 2.5 mM EDTA, and 0.5% NP-40 for 5 min at 4°C. ATP was measured with the ATP determination kit (Molecular Probes, Waltham, MA USA), according to the manufacturer's instructions. Experiments were performed in triplicate. Lactate secreted from hepatocytes into the medium was measured using the Lactate Assay Kit (Biovision). The average fluorescent intensity was calculated for each condition of replicates. Values were normalized to protein content obtained from the same wells.

**Fatty-acid oxidation.** FAO was assayed by quantifying the conversion of $^{14}$C-labeled palmitate to $^{14}CO_2$. In brief, AML12 cells were incubated in low-glucose DMEM containing 1 mM carnitine, 1% bovine serum albumin (BSA), 0.3 µCi/ml $^{14}$C-labeled palmitate (Perkin Elmer) for 4 h. FAO was determined by incubating culture media with an equal volume of 1 M perchloric acid. Samples were centrifuged at 2200 × $g$ at 4° C for 10 min. Released $^{14}CO_2$ was trapped in 1 N sodium hydroxide and the amount of radioactivity was then measured using a liquid scintillation counter (MicroBeta2 Microplate Counters, Perkin Elmer). After medium remotion, cells were lysed in 0.1 N NAOH and protein quantification was used to normalize FAO rate. Cells treated with 100 µM AICAR to stimulate FAO or with 100 µM Etomoxir to inhibit FAO were used as controls.

To analyze hepatic FAO in eIF6[+/+] and eIF6[+/−] mice, livers were quickly excised from anesthetized mice, placed in ice-cold isolation buffer (100 mM KCl, 40 mM Tris-HCl, 10 mM Tris-base, 5 mM MgCl$_2$-6H$_2$O, 1 mM EDTA, and 1 mM ATP; pH 7.4) and thoroughly minced in SET buffer (250 mM sucrose, 1 mM EDTA, 10 mM Tris-HCl, and 2 mM ATP; pH 7.4). Liver suspensions were then homogenized on ice with a Teflon pestle to lyse cells while keeping the mitochondria intact. Next, fresh liver extracts were incubated at 37° C for 1 h with the oxidation reaction mixture (0.3% BSA/100 µM palmitate/0.4 µCi/mL $^{14}$C-palmitate in 100 mM sucrose, 10 mM Tris-HCl, 10 mM K$_3$PO$_4$, 100 mM KCl, 1 mM MgCl$_2$-6H$_2$O, 1 mM ʟ-carnitine, 0.1 mM malate, 2 mM ATP, 0.05 mM CoA, and 1 mM DTT; pH 7.4). 70% perchloric acid was added to terminate all biological reactions. Upon perchloric acid addition, unoxidized bound palmitate precipitated from the solution while Acetyl-CoA remained soluble. The released $^{14}CO_2$ was trapped by NaOH. The amount of radioactivity in the captured $^{14}CO_2$ was then measured using a liquid scintillation counter. FAO rate was normalized to mouse body weight and represented in terms of captured $CO_2$ production rate. Before the addition of the radiolabeled substrate, the liver homogenate was incubated in the presence or absence of 1 mM AICAR to stimulate FAO, or in the presence or absence of 10 µM rotenone to inhibit mitochondrial β-oxidation.

**Fatty-acid synthesis.** Following a 20 h incubation in serum-free low-glucose DMEM, FAS was measured on AML12 cell lines. Fresh low-glucose medium containing 4 μCi/mL D-[6-$^{14}$C]-glucose (Perkin Elmer) was added for 4 h. Cells were washed twice with D-PBS before lysis in 0.5% Triton X-100. The lipid fraction was extracted by sequential addition of chloroform and methanol (2:1 v/v)[14]. After washing with water, phase separation was achieved by low-speed centrifugation (1000 × $g$, 15 min), and $^{14}$C incorporation into the lower lipid-containing phase was counted using a liquid scintillation counter (MicroBeta2 Microplate Counters, Perkin Elmer). Each cell line was assayed in triplicate, and readings were normalized to total cellular proteins (μg).

**Immunofluorescence.** Immunofluorescence was performed on primary hepatocytes as previously described[10]. Two different primary antibodies were used, Tomm20 (Cell Signaling, Cat#42406) and AIF (Cell Signaling, Cat#5318), and incubated overnight at 4 °C. Alexa-Fluor 555-conjugated secondary antibodies (1:500, Thermo Fisher Scientific) were added for 1 h at room temperature. Slides were mounted in Mowiol 4-88 mounting medium (Sigma-Aldrich). Images were acquired using a filterless laser-scanning confocal microscope (Leica TCS SP5) using excitation wavelengths of 488 nm for green fluorescence (autofluorescence), 561 nm for red fluorescence (immunofluorescence), and 405 nm for DAPI (Sigma) nuclear labeling. Quantifications of Tomm20 and AIF fluorescence intensity per cell and per field of view were measured by processing raw data with NIS-Elements v.5.2 digital imaging analysis software (Nikon Instruments) for segmentation and precise quantification, implementing the general-analysis tool-box. AIF and Tomm20 mitochondrial fluorescence intensity levels were plotted as mean ± SEM values of each field of view. All the images were further processed with Photoshop CS6 (Adobe) software. For the statistical analysis of fluorescence intensity levels, the Mann–Whitney paired comparison was used.

**Translational analysis**

*Polysome profile analysis.* Liver samples were homogenized in 50 mM Tris-HCl, pH 7.8, 240 mM KCl, 10 mM MgSO$_4$, and 5 mM DTT, 250 mM sucrose, 2% Triton X-100, 90 μg/ml cycloheximide, and 30 U/ml RNasin using a glass douncer. Heparin (100 μg ml$^{-1}$) was added to the liver extract. After centrifugation at 12,000 × $g$ for 10 min at 4 °C, cytoplasmic extracts with equal amounts of RNA were loaded on a 15–50% sucrose gradient and centrifuged at 4 °C in an SW41Ti Beckman rotor for 3 h 30 min at 39,000 r.p.m. The gradient was analyzed by continuous flow absorbance at 254 nm, recorded by BioLogic LP software (Bio-Rad), and fractions were collected and kept on ice to avoid RNA degradation[10]. mRNA isolated from all considered fractions was used for real-time PCR analysis.

*Synthesis of plasmids, cell transfection, and luciferase assays.* The pRL YY1 5′-UTR plasmid was obtained by cloning the 5′-UTR of YY1 (ENSMUSG00000021264) transcript Yy1-201 (ENSMUST00000021692.8) at the Nhe1 site of the Renilla Luciferase reporter (pRL) plasmid. For transient transfection, cells were seeded into six-well plates, 24 h before transfection. HEK293 cells were transfected with either the pRL or the pRL YY1 5′-UTR plasmid using calcium phosphate. After 24 h, the medium was changed with fresh DMEM supplemented with 100 nM rapamycin or 0.05% dimethyl sulfoxide, as a control, for 4 h. Renilla Luciferase activity was detected using the DualGlo Luciferase System (Promega) and values were normalized on Renilla Firefly mRNA abundance using SYBR green oligos as indicated. RenFW: 5′-GGAATTATAATGCTTATCTACGTGC-3′ and RenRW: 5′-CTTGCG AAAAATGAAGACCTTTTAC-3′. Experiments were run in triplicate.

**Western blotting.** Western blotting analysis was performed on protein extracts obtained from liver homogenized in radioimmunoprecipitation assay buffer (10 mM Tris-HCl, ph 7.4, 1% sodium deoxycholate, 1% Triton X-100. 0,1% SDS, 150 mM NaCl and 1 mM EDTA, pH 8.0). The quantitation of the relative amount of YY1 and C/EBPβ-LIP proteins was obtained by densitometric analysis using the ImageJ Software.

**Statistics and reproducibility.** Each experiment was repeated at least three times, as biological replicates; means and standard deviations between different experiments were calculated. Student $t$ test was used and statistical $P$ values obtained were indicated either as numbers or as asterisks: ****$P$ values <0.0001, ***$P$ values <0.001, ** for $P$ values <0.01, and *$P$ values <0.05. Experimental numbers with animal samples were subject to power calculations. For KM survival curves of primary liver cancer patients, log-rank $P$ value was used for statistical significance. Microphotographs are representative of at least 50 fields (1b, 3g, h, 5g, 8b, S5a, S5b, S6a). Western blots have been performed at least five times (5i, 7e, 7h, S1a, S1b, S4c, d, f, S6e). Bright-field images were taken >10 times (2b, 2e, 8b; S4e).

**Images.** Figures were drafted in PowerPoint 2016 (Microsoft) and finalized using Illustrator 2021 (Adobe). Some images were free and adapted from Servier Medical ART (https://smart.servier.com).

**Deposited sequences.** The RNA-seq data are available with accession number ID: E-MTAB-9009.

**Reporting summary.** Further information on research design is available in the Nature Research Reporting Summary linked to this article.

## Data availability

All data supporting the findings of this study are provided within the paper and its Supplementary information. We confirm that all mouse strains used here are readily available from the authors (eIF6 knockout) or from EMMA mouse repository (EM:07840). Further information on research design is available in the Nature Research Reporting Summary linked to this article. Any data are available from the authors upon request. The RNA-seq data generated in this study corresponding to Supplementary Data 1 have been deposited in the www.ebi.ac.uk/arrayexpress/ database under accession code ID: E-MTAB-9009. Retrieved human data were from GEO, https://www.ncbi.nlm.nih.gov/sites/GDSbrowser?acc=GDS4881 (32 NAFLD). The data set of 125 obese patients, published in ref. [17] is under the jurisdiction of the co-author Luca Valenti who notifies that the ethical approval of the study does not allow to publicly share the patients' genetic data. All data, code, and materials used in the analysis are available upon reasonable request for collaborative studies regulated by materials/data transfer agreements (MTA/DTAs) to the corresponding author (luca.valenti@unimi.it). Source data are provided with this paper.

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

## Acknowledgements

We acknowledge Fabrizio Loreni for helpful and enlightening discussions who sadly passed away while the manuscript was in preparation. We thank Dr. Mario Tirone for helpful suggestions, Dr. Roberta Alfieri for preliminary bioinformatics analysis, and Dr. Riccardo Rossi for bioinformatics support. We sincerely thank C.F. Calkhoven and C. Muller for kindly providing us C/EBPβ construct vectors and helpful scientific discussion and Scott Friedman lab for help with the NASH-HCC mouse model. This work was supported by ERC Grant Translate 338999, AIRC IG 2017, and Fondazione Invernizzi contribution to S.B. Finally, we are thankful for the support we have received from CNCCS s.c.a.r.l for the development and production of eIF6 inhibitors.

## Author contributions

A.S., A.M and S.B. designed the study. A.S and A.M carried out the murine colony management, conducted experiments, and analyzed the data. G.B. performed the transcriptomic analyses of human liver biopsies. L.V., R.D. and M.M contributed to human sample collection and provided clinical information. G.V. performed all computational analysis on murine samples. G.V., A.S. and A.M. analyzed the RNA-seq data. S.B. and G.V. mined data and performed the statistical analysis. C.C. performed Immuno-fluorescence images acquisition and statistical analysis. D.C. helped in Real-Time PCR analysis. S.R. analyzed metabolic data and provided critical discussions. D.B. helped in preliminary in vivo experiments. S.O. helped in Immunostainings and computational analysis. N.M. performed Synthesis of Plasmids, Cell Transfection, and Luciferase Assays experiments. A.B. provided us with chemical compounds to inhibit eIF6. A.S., A.M. and S.B. discussed the results and wrote the paper.

## Competing interests

The authors declare no competing interests.
