## [Peer Review File · Nature Communications]

Reviewers' comments:

Reviewer #1 (Remarks to the Author):

In this manuscript, Scagliola and colleagues show that eIF6 reduction decreases hepatic steatosis in mice subjected to high fat diet. This effect might be mediated by translational regulation of the transcription factor C/EBP-beta by eIF6. Additionally, eIF6 heterozygous mice show increased translation of YY1, which may explain increased mitochondrial activity (and perhaps contribute to counteracting oncogenic transformation). Interestingly, two new inhibitors of eIF6 are tested in vitro. When tested on hepatic AML12 cells, these inhibitors lead to lesser lipid accumulation, lesser translation C/EBP-beta, and increased translation of YY1. The inhibitors, thus, mimic the effects of eIF6 reduction in heterozygous mice.

In general, the manuscript is interesting and increases our understanding of the role of eIF6 in lipid accumulation and mitochondrial activity. However, some results are not novel and have been already published in the previous Nat Commun paper from this lab (Brina et al 2015). In addition, the role of the eIF6-YY1 axis in carcinogenesis is not explored. The manuscript will benefit from additional experiments to ensure a causal relationship between eIF6-mediated regulation of C/EBP-beta and fatty acid metabolism, as well as of YY1 and mitochondrial metabolism. Some re-writing should also be done to avoid over-statements.

Major comments:

1) That eIF6 regulates lipid metabolism and C/EBP-beta expression has already been published by these authors (Nat Commun 2015). In this previous manuscript, reduced body weight of eIF6 +/- mice upon high-fat diet feeding, and reduced levels of triglycerides and cholesterol are shown (data have been repeated and correspond to Figs 3a-c of the current manuscript). Levels of ATP and lactate (Fig 5f in this manuscript) were also measured in the previous paper, but curiously the results are different. While here increased ATP levels and no change in lactate secretion are observed in eIF6 +/- mice, in the Nat Commun 2015 paper these authors showed decreased ATP levels and decreased lactate secretion. Can the authors explain the reasons for these differences?

2) The title is over-stated. The authors have targeted eIF6 (not translation; targeting translation by other means may have different effects) and have not really shown that 'targeting eIF6 reduces NAFLD and prevents activation of carcinogenesis', as this would have required the use of their inhibitors to block eIF6 function and a profound analysis of carcinogenesis. They have actually shown that mice with reduced levels of eIF6 (heterozygous mice) have decreased NAFLD. No data on carcinogenesis has been included with the exception of clonogenic assays in cultured cells. Therefore, I strongly suggest to change the title to more accurately reflect the conclusions.

3) Figure 1b,c: Please, give details on the second patient cohort used for IHC staining, as well as details on methods (antigen retrieval, antibody refs and dilutions, etc). Show quantifications of the whole cohort in addition to the single images currently included in the figure.

4) Figure 2b: Data on cellular transformation by eIF6 depletion are not convincing. Only a clonogenic assay is shown, where the quality of the image is poor. Please, show a better image, and measure cellular transformation using alternative assays (3D-growth, colony formation on agar, cell motility, anoikis resistance, etc). By the way, which cells are these, 293T or AML12? Transformation assays on AML12 liver cells are recommended, given that this manuscript is centered on the liver.

5) Figure 4a-c: Genes in Fig 4a are not the same as those in Fig 4b-c. Why is that? Genes in parts b-c should appear in part a, as this is an unbiased clustering.

6) Figure 4: A Western blot of C/EBP-beta in livers from eIF6+/+ vs eIF6+/- mice is required to show that the levels of this transcription factor are indeed reduced.

7) To claim that C/EBP-beta is the mechanistic link between eIF6 and fatty acid metabolism, some

experimental proof should be provided. For example, does over-expression of C/EBP-beta rescue the effects of eIF6 depletion on adipogenesis in liver cells?

8) Lanes 179-180: The results on Ppar (alpha and gamma) are not in line with the effects of eIF6 deficiency, as the authors state, but are actually contradictory. If the aforementioned genes have a protective effect in NAFLD-NASH progression and eIF6 deficiency reduces the levels of those genes, then eIF6 deficiency should promote NAFLD-NASH progression; however, the opposite is observed. This contradiction is recognized by the authors in the discussion, and it is misleading that the authors state that the data are 'in line' in the results section.

9) Lanes 195-197: eIF6 mRNA is not always over-expressed during HCC progression, as stated. According to Table S1, this is so in mouse models, but not in patient samples, where expression of eIF6 is higher in tumors with no metastasis compared with tumors with venous metastasis.

10) Figure 4e and lanes 199-200: The authors mention that 'HCC-predictive consensus signatures perfectly overlap with the signature of genes inhibited by eIF6 depletion'. What does this exactly mean? Which is this signature and how many genes overlap? Please, indicate in Figure 4e.

11) Figure S6a and lanes 239-240: The authors state that eIF6 wt and heterozygous mice had similar levels of polysomes, but the figure shows higher levels of heavy polysomes (as recognized by the authors in the figure legend), consistent with the higher phosphorylation levels of RPS6 and 4E-BP1. This would lead to increased global translation. How can the authors explain increased global translation under reduced eIF6 levels? Can the authors measure global translation metabolically to confirm the predictions from the polysome profile?

12) Figure 6: Does mitochondrial activity in eIF6 heterozygous cells decrease after YY1 depletion? This experiment is necessary to ensure a functional relationship between mitochondrial fitness and YY1, as stated in the title of the corresponding section (lanes 209-211) and the figure legend.

13) Do the eIF6 inhibitors described here have an effect in preventing or ameliorating steatosis in eIF6^{+/+} mice subjected to high fat diet? I understand that this might be a laborious experiment, but it would dramatically increase the value of the manuscript.

14) Supplementary Excel Table: Please, add columns showing the average counts per gene and the statistical significance of differences between wt and heterozygous mice. Show also gene names.

15) In the discussion, while I very much like the concept of 'metabolic learning', the authors insist on a carcinogenic branch (progression to HCC) that has not been explored in this manuscript and remains speculation. Therefore, they cannot state 'this is indeed what happens' when talking about HCC. The authors cannot also say that 'YY1 is the last step of a translation pathway that requires mTORC1 and eIF4F', as they have not proven that YY1 is the critical factor that mediates the increase of mitochondrial activity in eIF6^{+/-} mice.

Minor comments:

- Figure 1e: Please, indicate the p-value, hazard ratio and confidence intervals.
- Please, explain AICAR in Figure 2h legend or in main text.
- Figure 3b can be eliminated, as it is redundant with Figure 3a.
- Figure 4a: why are Egfr, Snip1 and Uqf3b in the cytoplasmic translation cluster?
- Figure 5a and S3b: Please, check sign of p-values (if significant they should all be positive in a -log scale).
- Overstatement, lane 128: 'eIF6 depletion prevents lipid synthesis'. Figure 2d actually just shows a mild reduction. Furthermore, as far as I understand, Oil Red O measures lipid content, not lipid synthesis.

Reviewer #2 (Remarks to the Author):

The manuscript entitled "Targeting of Translation Induces a Metabolic Rewiring that Reduces NAFLD and Prevents Activation of Carcinogenesis" presents interesting data on the role of eIF6 on transcriptional/translational regulation of liver metabolism under high fat diet (HFD). The authors propose that eIF6 targeting have a positive effect on hepatocytes and liver metabolism preventing fat liver disease and eventually cancer. The manuscript shows a positive correlation between eIF6 expression and human NAFLD, cirrhosis and liver cancer. Gene expression analysis of livers from a mouse model with reduced eIF6 (eIF6 +/-) under HFD shows a reduction of lipid accumulation and lipogenesis-related genes, and an increased mitochondrial bioenergetics. The authors present a strong genetic and GO analysis on the effect of eIF6 on lipogenic-related factors, and describe an mTORC1-dependent mechanism for the increased mitochondrial activity in eIF6 +/-, which could support their observation on FAO. Finally the authors present two pharmacological eIF6 inhibitors as prospective drugs to treat liver disease.

The amount of data is very high, and the paper is resented very clearly. Unfortunately, the manuscript presents a number of serious issues that precludes its publication in its present status.

Contrary to the manuscript title, the prevention of carcinogenesis activation by targeting protein translation is not properly proven in the manuscript, are most of the results are merely correlative. The work fails to demonstrate that eIF6 is a key factor for the transition from NAFLD to HCC, and the causality of eIF6 in NAFLD to HCC transition was not established in this work.

A second issue is that the novelty of the manuscript is questionable since the effect of eIF6 depletion on HFD-induced obesity, insulin resistance and steatosis has been already published by the same lab (Brina et al 2015 NatComm). In this previous publication, the authors already showed the relevance of eIF6 on lipogenesis translation/transcriptional program. Sadly, whole liver pictures shown in Figure 3e were already published in the 2015 paper, a fact that somehow reflects a reduced novelty of the current manuscript.

Considering the previous points, and since the present manuscript is aimed at focusing on the role of eIF6 driving liver disease under HFD condition, more specific experiments proving the effect of eIF6 depletion on NAFLD progression and hepatocyte transformation and tumorigenesis should be included.

Other points, with no particular order of priority:

- Fig. 2d: Oil Red O staining changes do not prove that lipid synthesis is inhibited, as claimed in the text (line 127). Higher consumption of newly synthesized lipids, as shown for the FAO assay using isolated livers (Fig. 2h), could alternatively explain the reduced Oil Red O staining.

- Fig. 2e: The difference between Sh Scramble-PA+ and Sh eIF6-PA+ cannot be compared statistically since they have been normalized to different controls.

- Fig. 2e-h: The AML12 experiments should include the FAO analysis, as in Fig. 2h with primary hepatocytes. On the other hand, since FAO analysis is performed on hepatocytes from the mouse model, it should be shown whether hepatocytes eIF6 +/- have a reduced palmitate-accumulation level compared to eIF6 +/+. Protein levels of eIF6 should be shown to confirm less eIF6 expression in eIF6 +/-.

- Line 164: The authors conclude that "eIF6 inhibition reduces de novo lipogenesis acting on C/EBPb translation". However, the results shown in Fig 4 and Sup Fig 3 - 4 are merely correlative, and do not prove a causality relationship. Indeed, these results show a positive correlation between eIF6, C/EBPb and lipogenic targets at the translational level and GO analysis. C/EBPb rescue experiments in eIF6 +/- hepatocytes would support a causal effect between both proteins. If C/EBPb is regulated at translational level, protein levels by immunodetection should be shown.

- Fig. 5a. GO analysis showing a downregulation of FAO-related pathways is in contradiction with the results obtained previously in Fig. 2e-h, and with the later results obtain in Fig c-d.

- Fig. 6 and Sup Fig. 6. To prove a mechanistic role of mTORC1 in the regulation of YY1 by eIF6, it would be important to confirm that rapamycin affects YY1 translation (expression) and mitochondrial bioenergetics on eIF6 +/- eIF6 +/+ and hepatocytes. Does rapamycin treatment decrease FAO?

- The beneficial effect of the eIF6 inhibitors on liver lipid metabolism (increased oxidation and reduced lipogenesis) to prevent NAFLD requires further confirmation. Primary obese hepatocytes (eIF6 +/-) should be treated with these inhibitors to check lipid accumulation, FAO, lipogenic-related genes, mitochondria activity, C/EBPb protein level, mTOR signaling, etc.

Reviewer #3 (Remarks to the Author):

Scagliola et al. provide convincing evidence supporting the role of translation mediated by eIF6 in NAFLD. The expression of eIF6 is increased in NAFLD as well as in the progression to hepatocellular carcinoma, where is associated with worse prognosis. Depletion of eIF6 is able to restrict tumorigenesis in vitro as well as lipid accumulation and fatty acid oxidation. Most importantly, eIF6 deficiency also decreases liver fibrosis and steatosis in vivo.

The study is nicely written and the discussion and interpretation of the results is rigorous. The study is clinically relevant and mechanistically interesting. There are some minor concerns that need to be addressed:

1. The link to carcinogenesis is not too strong (only in vitro data) so I think it would be more appropriate to remove from the title.
2. In figure 1D, is there a statistical test performed? It seems like only hepatocytes have higher levels of eIF6 and not other cells? Please clarify.
3. It would be useful to include in the text (not only in methods) the cell line that is used in the experiments (Fig. 2), indicating the features of the cell line (immortalized hepatocytes from mice).
4. It would be important to repeat key experiments in Fig. 2 using at least a second shRNA as RNAi is known to have off-target effects.

GENERAL ISSUES

We have produced data employing a novel mouse model for studying the evolution from NASH to hepatocellular carcinoma¹. We now demonstrate that the prediction that eIF6 inhibition impaired carcinogenesis, *in vivo*, was right (new Figure 5). We have modified the title to “*Targeting of eIF6-Driven Translation Induces a Metabolic Rewiring That Reduces NAFLD and the Consequent Evolution to Hepatocellular Carcinoma*” which faithfully reflects the scientific content of the paper.

The issue on novelty and contradiction with previous data has been explained. There is no contradiction with previously published data. The results were different because the experiments were conceptually different. As a consequence, the findings are novel.

Experiments on fatty acid synthesis, fatty acid oxidation have been added.

All specific points are addressed below.

Reviewers' comments:

Reviewer #1 (Remarks to the Author):

(...) In general, the manuscript is interesting and increases our understanding of the role of eIF6 in lipid accumulation and mitochondrial activity. However, some results are not novel and have been already published in the previous Nat Commun paper from this lab (Brina et al 2015). In addition, the role of the eIF6-YY1 axis in carcinogenesis is not explored. The manuscript will benefit from additional experiments (...). Some re-writing should also be done to avoid over-statements.

We have addressed the points raised. Please see the specific responses in the point-by-point section.

Major comments:

1) That eIF6 regulates lipid metabolism and C/EBP-beta expression has already been published by these authors (Nat Commun 2015). In this previous manuscript, reduced body weight of eIF6 +/- mice upon high-fat diet feeding, and reduced levels of triglycerides and cholesterol are shown (data have been repeated and correspond to Figs 3a-c of the current manuscript). Levels of ATP and lactate (Fig 5f in this manuscript) were also measured in the previous paper, but curiously the results are different. While here increased ATP levels and no change in lactate secretion are observed in eIF6 +/- mice, in the Nat Commun 2015 paper these authors showed decreased ATP levels and decreased lactate secretion. Can the authors explain the reasons for these differences?

The best way to answer this criticism is starting from the point raised by “*Levels of ATP and lactate (Fig 5f in this manuscript) were also measured in the previous paper, but curiously the results are different*”. The alleged experiments in the two papers were very different, and gave different results. In Brina et al.¹, the metabolites were measured postprandially, in normal mice. Here, these metabolites were measured at the basal level, after the insulting effects of a long term HFD. The observed difference is therefore an important result (see next paragraph). The reviewer is right that we were not clear on this point, now done, page 11, from line 262, “After the HFD regimen...”. The new Figure is Figure 6f-g.

From there everything becomes consequential. First, the results are not ad odds, they are different because the experiments are different. Second, these results underline the role of eIF6 in the temporal evolution of the disease. In short: eIF6 higher levels, in a healthy liver, postprandially, increase lactate, lipogenesis, ATP¹. This

paper: the HFD challenges the eIF6^{+/+} liver exacerbating lipid accumulation, whereas the chronic reduction of eIF6 protects the liver from damage. Consequently, the liver of wt mice in the chronic HFD is more damaged, has less mitochondria, and less ATP. In contrast, the eIF6-depleted liver is protected, is healthier and has more ATP. Another way to see our findings is that “higher glycolytic and translation rate permitted by postprandial eIF6 activity, associated with HFD meals, progressively damage the liver”.

The fact that high eIF6, in the long run, damages the liver more than eIF6 reduction” is shown also by new Supplementary Figure 6, panels a-c, page 11, from line 287. Here we show the evolution of liver translation, analysed by puromycin incorporation. At early HFD, the wt liver translates more because eIF6 is rate-limiting. With increasing damage, the wt liver reduces translation, whereas the eIF6-depleted liver maintains protein synthesis (and may even show an increase that probably reflects a different evolutionary stage of the disease). These findings are in agreement with our model that increased translation enforces HFD damage and HFD damage then reduces translational capability in a feed-forward mechanism. Page 14, line 361 discusses the data.

2) The title is over-stated. The authors have targeted eIF6 (not translation; targeting translation by other means may have different effects) and have not really shown that ‘targeting eIF6 reduces NAFLD and prevents activation of carcinogenesis’, as this would have required the use of their inhibitors to block eIF6 function and a profound analysis of carcinogenesis. They have actually shown that mice with reduced levels of eIF6 (heterozygous mice) have decreased NAFLD. No data on carcinogenesis has been included with the exception of clonogenic assays in cultured cells. Therefore, I strongly suggest to change the title to more accurately reflect the conclusions.

We changed the title. Most importantly, however, we added data. We added new Figure 5 (from page 9, line 229) that shows that tumorigenesis and tumor growth are delayed in conditions of eIF6 depletion. Figure 5 is derived from a new experiment in which we performed high-fat diet plus high-sugars drink water, combined with low weekly dose of intraperitoneal carbon tetrachloride (CCl₄) to induce HCC, as recently shown by Friedman’s group¹. A mere 50% eIF6 reduction has an impact in the number and size of nodules, even in this highly tumorigenic model (100% penetrance in six months).

3) Figure 1b,c: Please, give details on the second patient cohort used for IHC staining, as well as details on methods (antigen retrieval, antibody refs and dilutions, etc). Show quantifications of the whole cohort in addition to the single images currently included in the figure.

The experimental details of the cohort² are included in the paper, page 15, line 395. The details of the immunohistochemistry have been added at page 20, line 530. We have added quantitation of pS6 staining (new Figure 1d) as requested; the immunohistochemistry quantitation was based on N=12 patients selected for each class that was diagnosed according to a histopathological scale³. Information has been added.

4) Figure 2b: Data on cellular transformation by eIF6 depletion are not convincing. Only a clonogenic assay is shown, where the quality of the image is poor. Please, show a better image, and measure cellular transformation using alternative assays (3D-growth, colony formation on agar, cell motility, anoikis resistance, etc). By the way, which cells are these, 293T or AML12? Transformation assays on AML12 liver cells are recommended, given that this manuscript is centered on the liver.

We have included the requested data on the AML12 hepatocyte cell line (Fig. 2a-c) in which we show number of colonies and their representative size (page 6). In addition, we performed the *in vivo* experiments (new Figure 5). eIF6 depletion reduces Myc-induced lymphomagenesis in mice⁴ and soft agar colony formation in MEFs transduced with dnp53/Myc-RAS⁵. eIF6 role in tumorigenesis is very consistent. Given the role of eIF6 also in translational control of lipid metabolism, we think that this paper adds a very important concept on how eIF6 modulates NASH evolution to cancer.

5) Figure 4a-c: Genes in Fig 4a are not the same as those in Fig 4b-c. Why is that? Genes in parts b-c should appear in part a, as this is an unbiased clustering.

We added it back (it was simply due to the fact that we thought it was redundant information and we had eliminated it, Figure 4).

6) Figure 4: A Western blot of C/EBP-beta in livers from eIF6+/+ vs eIF6+/- mice is required to show that the levels of this transcription factor are indeed reduced.

According to the uORF-model of translational regulation driven by eIF6, the LIP isoform of C/EBP-beta should be reduced. We have included these data in Supplementary Figure 4c (representative blot and N=3 quantitation), showing and quantifying the observed reduction of LIP in eIF6^{+/-} mice (page 8, from line 199).

7) To claim that C/EBP-beta is the mechanistic link between eIF6 and fatty acid metabolism, some experimental proof should be provided. For example, does over-expression of C/EBP-beta rescue the effects of eIF6 depletion on adipogenesis in liver cells?

According to genetic evidence the LIP expression is important for C/EBP-beta activity⁶. New supplementary Figure 4 (d-g) shows that LIP overexpression rescues the lipid accumulation reduction driven by eIF6 depletion (page 8, from line 202). IMPORTANT. Please note that we do not claim that eIF6 function is all exerted through C/EBP-beta and, viceversa, all C/EBP-beta is regulated through eIF6, as the whole landscape of gene expression is coordinately changed by eIF6 (page 15 from line 370; Source Data, Excel File_Normalized Counts). In this respect, we think that mRNAs with uORF that regulate metabolism, glycolytic and lipogenesis, co-evolved with eIF6 activity.

8) Lanes 179-180: The results on Ppar (alpha and gamma) are not in line with the effects of eIF6 deficiency, as the authors state, but are actually contradictory. If the aforementioned genes have a protective effect in NAFLD-NASH progression and eIF6 deficiency reduces the levels of those genes, then eIF6 deficiency should promote NAFLD-NASH progression; however, the opposite is observed. This contradiction is recognized by the authors in the discussion, and it is misleading that the authors state that the data are 'in line' in the results section.

It is a fact that Ppara and Pparg are reduced in the eIF6 knockout mice, at the mRNA level. It is a fact that several lines of research suggest that Pparg agonists “reduce” NAFLD-NASH, whereas some other lines of evidence suggest that Pparg increases steatosis (reviewed in⁷, and, for instance⁸). However, several interpretations are possible at this stage, for our data. We have commented the results in the discussion, page 15, from line 370 and we have corrected the misleading sentence (deleting in line).

9) Lanes 195-197: eIF6 mRNA is not always over-expressed during HCC progression, as stated. According to Table S1, this is so in mouse models, but not in patient samples, where expression of eIF6 is higher in tumors with no metastasis compared with tumors with venous metastasis.

This was an embarrassing mistake. We apologize. In the submission, we inserted a Table (correctly spotted by this reviewer) that was not supposed to be in the article. The table was not specifically referred to, since in that section we were discussing mice models, and was even wrongly edited. I will explain why the “alleged” Table does not challenge our conclusions and can be removed. The Table that was inserted, is below:

Patients	GEO Profile Accession number	eIF6 mRNA expression		P value
		No metast. (mean ± SD)	Venous metast. (mean ± SD)	
HCC	GDS3091	0,4161 ± 0,20	0,1228 ± 0,21	0,01712

Now: 1. The GDS3091 study that is mentioned is on “**Noncancerous** hepatic tissues from hepatocellular carcinoma patients with and without venous metastasis”. It was a study on the “microenvironment”. 2. The value in front of **No metast.** was -04161 (not + but negative). This minus was lost in a typo. 3. A GEO figure from those data can be rescued, is pasted aside, and shows that the signal quality is variable (dataset is available at <https://www.ncbi.nlm.nih.gov/geo/profiles/48365949>). In short, neither the results change our work, nor they belong here. Why a wrongly edited and misplaced table went there, and was not spotted by all the editing authors is unclear. Apparently, we printed to .pdf a wrong file that was added in the n-version of the revisions and we missed it.

This said: human mRNAseq/microarray studies consistently see higher expression of eIF6 in HCC compared to surrounding tissue. For instance, of the ones we data-mined 8/9 showed statistically significant increased expression (with P-values ranging from 0.007 to $2e^{-79}$). In other terms humans and mice are very similar. However, the table has been deleted and will be not substituted since this section is mouse-specific. Sorry for the mistake.

10) Figure 4e and lanes 199-200: The authors mention that ‘HCC-predictive consensus signatures perfectly overlap with the signature of genes inhibited by eIF6 depletion’. What does this exactly mean? Which is this signature and how many genes overlap? Please, indicate in Figure 4e.

We changed the Figure and the text. The “HCC-predictive consensus signature” referred to a metadata analysis that was correctly cited in the original version as ref. 28 and concluded that a signature of 25 “lipid-related genes” was altered in the progression from NASH to HCC⁹, in humans. Our statement meant that the genes described in that paper, when expressed also in mice, were consistently altered in our dataset (old Figure 4e/f). We deleted old 4e and 4f and included a Table with 18 key-genes (Supplementary Table 3). Full list of differential expressed genes which correlate to severe NAFLD has been inserted in a new Excel File Human data sheet 4. We have rephrased, because we agree with this reviewer that does not make much sense to imply the existence of an accepted and unique HCC-predictive consensus signature (from page 9, line 210, first paragraph). In reality, there are many signatures of liver disease which show similarities to the eIF6 knockout. We have inserted another one in new Fig. 5a from a more recent analysis¹⁰. The details are shown in Supplementary Table 2 and in the Source Data, Excel File Human data sheets 1-3. Finally, we have validated some of the predictive genes in new Fig. 5j, from the new HCC model that we built. The point that we want to make is that eIF6-driven gene expression predicts what we (now) observe in the NASH-HCC model.

11) Figure S6a and lanes 239-240: The authors state that eIF6 wt and heterozygous mice had similar levels of polysomes, but the figure shows higher levels of heavy polysomes (as recognized by the authors in the figure legend), consistent with the higher phosphorylation levels of RPS6 and 4E-BP1. This would lead to increased global translation. How can the authors explain increased global translation under reduced eIF6 levels? Can the authors measure global translation metabolically to confirm the predictions from the polysome profile?

These are relevant observations. We did not want to “stress” the modest but reproducible increase in polysomes that is in line with the higher phosphorylation levels of RPS6. A number of experiments and changes have been

made to address this point. First, we performed the requested experiments of metabolic labelling (Supplementary Figure 6). Results: higher eIF6 level at the beginning of HFD increase insulin-stimulated translation and lipogenesis. In the HFD setup, the positive lipogenic feedback given by eIF6, damages the liver. Damaged livers progressively translate less and have less pS6 stimulation (probably due to insulin resistance). Eventually, in the long term, eIF6 knockout livers that are not subject to the positive lipogenic feedback mediated by eIF6, are less damaged and translate more. All this work is in new Supplementary Figure 6 and explained from page 11, line 284. In brief, we provide the logic explanation on *"How can the authors explain increased global translation under reduced eIF6 levels"*. After HFD eIF6 knockouts are healthier than wt mice and translate more, in spite of having less eIF6, because mTORC1 translational branch is active. Last, concerning the polysome data, we substituted the word similar with higher, where we indeed meant not dramatically different (page 11, line 27). We discussed the data (page 14, line 361).

12) Figure 6: Does mitochondrial activity in eIF6 heterozygous cells decrease after YY1 depletion? This experiment is necessary to ensure a functional relationship between mitochondrial fitness and YY1, as stated in the title of the corresponding section (lanes 209-211) and the figure legend.

To answer this point and also in response to a suggestion of reviewer 2, we have verified that increased (mitochondrial) fatty acid oxidation, due to eIF6 downregulation, is indeed reduced by YY1 depletion, with two shRNAs (**Figure 7h, i**). Data are described at page 12, from line 298.

13) Do the eIF6 inhibitors described here have an effect in preventing or ameliorating steatosis in eIF6+/- mice subjected to high fat diet? I understand that this might be a laborious experiment, but it would dramatically increase the value of the manuscript.

The amount of the compounds that we have, do not allow the experiment *in vivo* on a sufficient number of mice. In addition, they must be tested for toxic effects, bioavailability and so on. An entire new project (and grant) is requested for this task, in order to obtain really sound data. The problem is defined in the last part of the discussion (page 15, line 385).

14) Supplementary Excel Table: Please, add columns showing the average counts per gene and the statistical significance of differences between wt and heterozygous mice. Show also gene names.

Done.

15) In the discussion, while I very much like the concept of 'metabolic learning', the authors insist on a carcinogenic branch (progression to HCC) that has not been explored in this manuscript and remains speculation. Therefore, they cannot state 'this is indeed what happens' when talking about HCC. The authors cannot also say that 'YY1 is the last step of a translation pathway that requires mTORC1 and eIF4F', as they have not proven that YY1 is the critical factor that mediates the increase of mitochondrial activity in eIF6+/- mice.

The connection to HCC is now less speculative (new Figure 5). The YY1 speculation has been removed. A general revision of the text has been done.

Minor comments:

- Figure 1e: Please, indicate the p-value, hazard ratio and confidence intervals.

Done

- **Please, explain AICAR in Figure 2h legend or in main text.**

Done (p. 12, line 301).

- **Figure 3b can be eliminated, as it is redundant with Figure 3a.**

3a described the global weight. Fig. 3b the gain of weight. We have corrected the x-axis definition of 3b that was wrong. Thanks for spotting the mistake (new Fig. 3).

- *Figure 4a: why are Egfr, Snip1 and Uqf3b in the cytoplasmic translation cluster?*

We relied on their classification in a GO dataset, without checking the actual function of these genes. Indeed, Snip1 is even a nuclear protein. We deleted them and notified the GO repository.

- *Figure 5a and S3b: Please, check sign of p-values (if significant they should all be positive in a -log scale).*

Indeed. We changed the figure orientation that gave a wrong impression (now Fig. 6a and Supplementary Fig.3b).

- *Overstatement, lane 128: 'eIF6 depletion prevents lipid synthesis'. Figure 2d actually just shows a mild reduction. Furthermore, as far as I understand, Oil Red O measures lipid content, not lipid synthesis.*

Corrected.

Reviewer #2 (Remarks to the Author):

The manuscript (omissis). The amount of data is very high, and the paper is resented very clearly. Unfortunately, the manuscript presents a number of serious issues that precludes its publication in its present status. Contrary to the manuscript title, the prevention of carcinogenesis activation by targeting protein translation is not properly proven in the manuscript, are most of the results are merely correlative. The work fails to demonstrate that eIF6 is a key factor for the transition from NAFLD to HCC, and the causality of eIF6 in NAFLD to HCC transition was not established in this work.

The reviewer was right that “*the prevention of carcinogenesis activation by targeting protein translation is (was) not properly proven in the manuscript*”. In new Figure 5 (page 9, line 229), we have performed *in vivo* experiments that show that eIF6 depletion reduces the transition from NASH to HCC, in a new mouse model¹. In addition, we substituted the words “prevent” with “reduce” that faithfully reflects our data. We wish to stress here that 50% eIF6 depletion is sufficient to reduce HCC burden in a mouse model that has 100% penetrance in 24 weeks! Thus, the gene expression prediction has been correct.

A second issue is that the novelty of the manuscript is questionable since the effect of eIF6 depletion on HFD-induced obesity, insulin resistance and steatosis has been already published by the same lab (Brina et al 2015 NatComm). In this previous publication, the authors already showed the relevance of eIF6 on lipogenesis translation/transcriptional program. Sadly, whole liver pictures shown in Figure 3e were already published in the 2015 paper, a fact that somehow reflects a reduced novelty of the current manuscript.

The images seem indeed identical even if 3e was not, according to our records, derived from the same k.o. mouse. Part of the issue is that eIF6-driven phenotypes are highly penetrant, and are independent from the genetic background. We changed them anyway (new Fig. 3e), because other readers may make similar conclusions. Thanks a lot for noticing.

Novelty. It is true, as this reviewer writes, that this work grounds on the work of Brina et al¹¹, where we showed that eIF6 regulated the translation of uORF-containing transcription factors for lipogenic pathways and reduced insulin resistance in a mild HFD model¹¹. The question is whether this paper is solid and has sufficient novelty. To our taste, we made a significant step forward showing that a) the progression NAFLD-NASH-HCC is reduced by eIF6 depletion (page 9), b) mechanistically, beside lipogenesis, a mitochondrial YY1

pathway is spared (page 10, from line 248), c) the eIF6 role in NASH-HCC progression seems identical in mice and humans (page 5), and d) eIF6 antagonists, *in vitro*, mimic the mechanistic action of eIF6 depletion (page 12, line 314). We believe that these findings are very important for the following reasons. Conceptually, as they identify an evident, evolutionarily conserved “feed-forward” mechanism in which eIF6-regulated translation exacerbates lipid-induced damage and links to cancer (page 13, from line 339). For therapy, because they show that, at least in principle, targeting eIF6 leads to effects that are much more specific than expected.

Considering the previous points, and since the present manuscript is aimed at focusing on the role of eIF6 driving liver disease under HFD condition, more specific experiments proving the effect of eIF6 depletion on NAFLD progression and hepatocyte transformation and tumorigenesis should be included.

We have inserted new Figure 5 with *in vivo* experiments including a NASH-HCC model from Friedman’s lab¹ where we show that a plain 50% depletion reduces the growth of HCC nodules (page 9, line 229). We have also inserted new transformation experiments in Figure 2a-c. (page 6, line 124). Other specific points are listed below.

Other points, with no particular order of priority:

- Fig. 2d: Oil Red O staining changes do not prove that lipid synthesis is inhibited, as claimed in the text (line 127). Higher consumption of newly synthesized lipids, as shown for the FAO assay using isolated livers (Fig. 2h), could alternatively explain the reduced Oil Red O staining.

True. The text has been deleted and new experiments have been performed to address the issue. New Fig. 2d shows that eIF6 depletion reduces FAS (measured by ¹⁴C-Glucose incorporation in FA), and also that it increases FAO, Fig. 2g (measured by ¹⁴C-palmitate oxidation to CO₂). Data are presented at page 6, from line 132.

- Fig. 2e: The difference between Sh Scramble-PA+ and Sh eIF6-PA+ cannot be compared statistically since they have been normalized to different controls.

First, a new shRNA has been added to have at least two. Second, the problem was graphic, not real. Indeed, we fix to 100% Sh Scramble-PA_{minus}, and all the values are compared to that point. The data are fully shown in Fig. 2f (page 6, line 135), with Fig. 2e showing the morphology.

- Fig. 2e-h: The AML12 experiments should include the FAO analysis, as in Fig. 2h with primary hepatocytes. On the other hand, since FAO analysis is performed on hepatocytes from the mouse model, it should be shown whether hepatocytes eIF6 +/- have a reduced palmitate-accumulation level compared to eIF6 +/-. Protein levels of eIF6 should be shown to confirm less eIF6 expression in eIF6 +/-.

Here three points are raised. FAO of AML12 has been done and is now in Fig. 2g. Acute palmitate accumulation in primary HFD hepatocytes was not done *in vitro*, because they are already quite full of it when we rescue them, but we have the *in vivo* analysis of FAO (Fig. 2h). The feasible experiment is now shown by *in vitro* analysis on AML12 (Fig. 2g). The requested blot is now in Figure 6c.

- Line 164: The authors conclude that “eIF6 inhibition reduces de novo lipogenesis acting on C/EBPb translation”. However, the results shown in Fig 4 and Sup Fig 3 - 4 are merely correlative, and do not prove a causality relationship. Indeed, these results show a positive correlation between eIF6, C/EBPb and lipogenic targets at the translational level and GO analysis. C/EBPb rescue experiments in eIF6 +/- hepatocytes would support a causal effect between both proteins. If C/EBPb is regulated at translational level, protein levels by immunodetection should be shown.

The reviewer is right on the “formal” lack of cause-effect data. Experiments have been added in new Supplementary Figure 4d-g, in AML12. The text is in page 8, from line 202. The overexpression of the LIP translation product of C/EBPb rescues, as predicted lipogenesis.

- Fig. 5a. GO analysis showing a downregulation of FAO-related pathways is in contradiction with the results obtained previously in Fig. 2e-h (Author Note: FAO increase), and with the later results obtain in Fig c-d.

Very intriguing point. We checked the GO dataset. This “incriminated” GO dataset contained only 5 genes: Alox12, Pparg, Pdk4, Ehhadh, Hadha, Irs1. Three of them are associated with specific steps of lipid oxidation (as for Pubmed analysis). All 5 unequivocally go down at the mRNA level in the eIF6 knockout (0.3-5-fold). However (measured) FAO increases in the eIF6 knockout. These are the facts. The reviewer is right that these “facts” are contradictory but since “*opposing things cannot be true*” we offer a plausible explanation. Possibly the “countertrend” changes of mRNA levels of these genes are not sufficient to impair FAO for whatever reason (e.g. they are only a subset; they are allosterically regulated; they are translated more, etc.) and simply demonstrate that translational regulation by eIF6 operates through an eIF6-specific mechanism. A reader can be however confounded by the presence of this result. We have taken the liberty to deal with the critic in this way. We have deleted the GO line, the p-value was low, since the set was made of only 5 genes (of course we deleted all the GO lines with \geq p-values).

, - Fig. 6 and Sup Fig. 6. To prove a mechanistic role of mTORC1 in the regulation of YY1 by eIF6, it would be important to confirm that rapamycin affects YY1 translation (expression) and mitochondrial bioenergetics on eIF6 +/- eIF6 +/- and hepatocytes. Does rapamycin treatment decrease FAO?

Very complex point. We do not claim “*a mechanistic role of mTORC1 in the regulation of YY1 by eIF6*” but that, upon HFD, cells that have reduced eIF6 levels have increased mTORC1 activity and higher YY1 expression regulated at the translational level. In short, if eIF6^{+/-} cells are, for instance, more insulin-responsive, they have higher mTORC1 activation, and therefore better mTORC1-regulated translation (eIF6 is not below mTORC1 pathway¹²).

Given these precisions, the mechanistic role of mTORC1 in the regulation of YY1 translation is demonstrated by the reporter assay of Supplementary Figures 6f-h performed by cloning the 5'UTR and measuring translation in the presence of a rapamycin treatment and internal controls (page 12, line 295). This assay is performed in a time-frame of hours and measures a very specific output, hence does not suffer from (evident) indirect effects of rapamycin.

RAPAMYCIN is a story by itself. The reviewer will notice that with the exception of the afore-mentioned short term and internally controlled experiment of Suppl. Figure 6f-h, rapamycin has not been considered in the project. There are conflicting reports on rapamycin and metabolism¹³. In general, rapamycin induces in the short term, *in vitro*, (most but not all reports/cells) an increase in FAO, autophagy, reduction of translation. However not all cells are sensitive, for instance ras mutations eliminate rapamycin sensitivity¹⁴. Besides, *in vivo*, rapamycin leads to paradoxical insulin resistance, and mitochondrial impairment. In this context, Sonenberg group showed that rapamycin reduces mitochondrial fitness at the translation level, indicating the importance of mitochondrial turnover effects^{15,16}. Our data are in line with the model that preservation of mTORC1 signalling is associated with mitochondrial integrity, also via YY1 mRNA translation which is not impaired by eIF6 depletion. Main Figure 8 and Supplementary Figure 8 also contribute to this model (page 13, line 326) since eIF6 “pharmacological” inhibition maintains active mTORC1 and YY1 translation.

- The beneficial effect of the eIF6 inhibitors on liver lipid metabolism (increased oxidation and reduced lipogenesis) to prevent NAFLD requires further confirmation. Primary obese hepatocytes (eIF6 +/-) should be treated with these inhibitors to check lipid accumulation, FAO, lipogenic-related genes, mitochondria activity, C/EBP β protein level, mTOR signaling, etc.

The experiments requested are **interesting, legitimate, and very important** to address and of course we will. However, after a 5-year long work, we need to establish firm points. The proof-of-concept that is required in this report is that these new eIF6 inhibitors affect the translation of the main target that is downstream of eIF6, C-EBP β but do not impair the translation of the target that is spared by eIF6 inhibition, YY1. In other terms

the inhibitors cause, in a simple model, the same effects as eIF6 depletion (Fig. 8). This said, we blotted for C/EBP β , YY1, P-4E-BP1 in AML12 cells treated with the inhibitors and we were excited to see that we confirmed the expected findings (page 13, line 326 and relative Figures). To treat obese hepatocytes, or even mice, and see if our inhibitors also “cure” the disease is the next frontier to cross! But even if these inhibitors fail *in vivo*, they will not disprove the concept that a) eIF6-driven translation exacerbates the transition from NASH to HCC, and b) inhibition of eIF6 activity reduces the lipogenic circuit, without affecting the mTORc1 cascade.

Final remarks, we hope that by showing the actual data on the NASH-HCC development, *in vivo*, the main criticism has been resolved.

Reviewer #3 (Remarks to the Author):

Scagliola et al. provide convincing evidence supporting the role of translation mediated by eIF6 in NAFLD. The expression of eIF6 is increased in NAFLD as well as in the progression to hepatocellular carcinoma, where is associated with worse prognosis. Depletion of eIF6 is able to restrict tumorigenesis in vitro as well as lipid accumulation and fatty acid oxidation. Most importantly, eIF6 deficiency also decreases liver fibrosis and steatosis in vivo. The study is nicely written and the discussion and interpretation of the results is rigorous. The study is clinically relevant and mechanistically interesting. There are some minor concerns that need to be addressed:

1. The link to carcinogenesis is not too strong (only in vitro data) so I think it would be more appropriate to remove from the title.

We have produced data employing a novel mouse model for studying the evolution from NASH to hepatocellular carcinoma¹. We now demonstrate that our prediction that eIF6 inhibition impairs the evolution from NAFLD to cancer was right. This said, considering suggestions from this reviewer, we will modify the title in “Targeting of eIF6-Driven Translation Induces a Metabolic Rewiring that Reduces NAFLD and the Consequent Evolution to Hepatocellular Carcinoma” that accurately reflects the data. New data are presented in **the novel Figure 5**.

2. In figure 1D, is there a statistical test performed? It seems like only hepatocytes have higher levels of eIF6 and not other cells? Please clarify.

The graph is a visual representation of single cell data retrieved from a published study that was cited¹⁷. The number of hepatocytes rescued in single cell experiment is low, as compared to infiltrating cells. In general, it is not common to include statistical analysis on these representations. We discussed the issue with the author who produced these data who proposed to apply traditional statistics to the raw data. To apply traditional statistics to a set of single cell data is not conventional, and our bioinformatics people discouraged us. In conclusion, the result shows that “on average” cirrhotic hepatocytes and cholangiocytes have higher eIF6 expression, it is visually intriguing and exploits current literature in a correct way. Following on this lead, we decided to challenge the hypothesis that *in vivo*, in our biopsies, we could see within the same individual that cholangiocytes in fibrotic areas had more eIF6 (**A**), and that hepatocytes in fibrotic areas had more eIF6 (**B**), when compared to “preserved” areas. We show two images below (for review only) that demonstrate that the hypothesis is correct. We also think on the basis of our data derived from¹⁷ it would have been more correct to write hepatocytes in cirrhotic regions, rather than cirrhotic hepatocytes. In conclusion, in a conservative fashion, we would like to leave the issue as it is.

A.

B.

eIF6 Immunohistochemistry followed by Hematoxylin counterstaining on human liver sections:

A. Enlargement of hepatic fibrotic area shows eIF6 abundance in cholangiocytes.

B. Liver section with different areas of hepatic damage shows diverse eIF6 expression in hepatocytes: low eIF6 (right image) and high eIF6 (left image) expression from two areas of the same liver (middle)

3. It would be useful to include in the text (not only in methods) the cell line that is used in the experiments (Fig. 2), indicating the features of the cell line (immortalized hepatocytes from mice).

Done. AML12 (page 6, line 124).

4. It would be important to repeat key experiments in Fig. 2 using at least a second shRNA as RNAi is known to have off-target effects.

True. A new shRNA has been added to have at least two. The data are fully shown in Fig. 2f (page 6, line 135). New experiments with YY1 shRNA have also been added and have been performed directly with two shRNAs, Figure 7h-i (page 12, line 302).

References

1. Tsuchida, T., *et al.* A simple diet- and chemical-induced murine NASH model with rapid progression of steatohepatitis, fibrosis and liver cancer. *J Hepatol* **69**, 385-395 (2018).
2. Baselli, G.A., *et al.* Liver transcriptomics highlights interleukin-32 as novel NAFLD-related cytokine and candidate biomarker. *Gut* **69**, 1855-1866 (2020).
3. Kleiner, D.E., *et al.* Design and validation of a histological scoring system for nonalcoholic fatty liver disease. *Hepatology* **41**, 1313-1321 (2005).
4. Miluzio, A., *et al.* Impairment of cytoplasmic eIF6 activity restricts lymphomagenesis and tumor progression without affecting normal growth. *Cancer Cell* **19**, 765-775 (2011).
5. Gandin, V., *et al.* Eukaryotic initiation factor 6 is rate-limiting in translation, growth and transformation. *Nature* **455**, 684-688 (2008).
6. Ackermann, T., *et al.* C/EBPbeta-LIP induces cancer-type metabolic reprogramming by regulating the let-7/LIN28B circuit in mice. *Commun Biol* **2**, 208 (2019).
7. Wang, Y., Nakajima, T., Gonzalez, F.J. & Tanaka, N. PPARs as Metabolic Regulators in the Liver: Lessons from Liver-Specific PPAR-Null Mice. *Int J Mol Sci* **21**(2020).
8. Lee, Y.J., *et al.* Nuclear receptor PPARgamma-regulated monoacylglycerol O-acyltransferase 1 (MGAT1) expression is responsible for the lipid accumulation in diet-induced hepatic steatosis. *Proc Natl Acad Sci U S A* **109**, 13656-13661 (2012).
9. Desterke, C. & Chiappini, F. Lipid Related Genes Altered in NASH Connect Inflammation in Liver Pathogenesis Progression to HCC: A Canonical Pathway. *Int J Mol Sci* **20**(2019).
10. Kaur, H., Bhalla, S., Kaur, D. & Raghava, G.P. CancerLivER: a database of liver cancer gene expression resources and biomarkers. *Database (Oxford)* **2020**(2020).
11. Brina, D., *et al.* eIF6 coordinates insulin sensitivity and lipid metabolism by coupling translation to transcription. *Nat Commun* **6**, 8261 (2015).
12. Ceci, M., *et al.* Release of eIF6 (p27BBP) from the 60S subunit allows 80S ribosome assembly. *Nature* **426**, 579-584 (2003).
13. Li, J., Kim, S.G. & Blenis, J. Rapamycin: one drug, many effects. *Cell Metab* **19**, 373-379 (2014).
14. Di Nicolantonio, F., *et al.* Deregulation of the PI3K and KRAS signaling pathways in human cancer cells determines their response to everolimus. *J Clin Invest* **120**, 2858-2866 (2010).
15. Morita, M., *et al.* mTORC1 controls mitochondrial activity and biogenesis through 4E-BP-dependent translational regulation. *Cell Metab* **18**, 698-711 (2013).
16. Morita, M., *et al.* mTOR Controls Mitochondrial Dynamics and Cell Survival via MTFP1. *Mol Cell* **67**, 922-935 e925 (2017).
17. Ramachandran, P., *et al.* Resolving the fibrotic niche of human liver cirrhosis at single-cell level. *Nature* **575**, 512-518 (2019).

Reviewers' comments:

Reviewer #1 (Remarks to the Author):

The authors have convincingly addressed my concerns, and the additional data included in the revised version make a strong case for eIF6-mediated translation as an important feature of adipogenesis during NAFLD and its progression towards HCC, with the identification of two functionally relevant downstream targets: YY1 and C/EBP β . I just have some minor corrections:

- Page 4, line 85: It is more correct to say 5' UTR rather than just 5', which refers to the very extreme of the mRNA.
- Page 5, heading: Rather than 'other ribosomal factors' it would be more correct to say 'other translation machinery factors', as initiation factors are not an integral part of the ribosome.
- Page 15, line 378: 'gene rearrangement' is normally used to indicate genes physically changing locations in the genome. I imagine the authors here refer to 'extensive transcriptomic changes'. Please, correct.
- Figure 1d: please, add to the figure legend n=12 per class, as stated in the rebuttal letter.
- The discussion is somewhat sketchy and could be better refined.

Reviewer #2 (Remarks to the Author):

The authors correctly replied to all my questions. I have no further comments.

Reviewer #3 (Remarks to the Author):

The authors have addressed the main points from the reviewers.

Response to Reviewers

Reviewer 1:

Page 4, lane 85: It is more correct to say 5' UTR rather than just 5', which refers to the very extreme of the mRNA.

True, corrected.

- Page 5, heading: Rather than 'other ribosomal factors' it would more correct to say 'other translation machinery factors', as initiation factors are not an integral part of the ribosome.

True, corrected.

- Page 15, lane 378: 'gene rearrangement' is normally used to indicate genes physically changing locations in the genome. I imagine the authors here refer to 'extensive transcriptomic changes'. Please, correct.

True, corrected.

- Figure 1d: please, add to the figure legend $n=12$ per class, as stated in the rebuttal letter.

True, corrected.

- The discussion is somewhat sketchy and could be better refined.

We made minor changes that can be tracked in the text. We thank this reviewer for his/her accuracy that helped to improve the manuscript.